# Gaussian Mixture autoencoder for uncertainty-aware damage identification in a Floating Offshore Wind Turbine

Ana Fernandez-Navamuel[1,2], Nicolas Gorostidi[2,3], David Pardo[3,2,4], Vincenzo Nava[5], and Eleni Chatzi[6]

[1]TECNALIA, Basque Research and Technology Alliance (BRTA), Parque Científico y Tecnológico de Bizkaia, Astondo bidea, Edificio 700, E- 48160 Derio, Spain
[2]Basque Center for Applied Mathematics (BCAM), Bilbao, Spain
[3]University of the Basque Country (UPV/EHU) Leioa, Spain
[4]Ikerbasque (Basque Foundation for Sciences), Bilbao, Spain
[5]Politecnico di Torino, Department DIATI, Corso Duca degli Abruzzi 24, 10129, Turin, Italy
[6]Institute of Structural Engineering, Department of Civil, Environmental & Geomatic Engineering, ETH Zurich, 8093 Zurich, Switzerland

**Correspondence:** Ana Fernandez-Navamuel (anafdeznavamuel@gmail.com)

**Abstract.** This work proposes an uncertainty-aware approach to the inverse problem of damage identification in a Floating Offshore Wind Turbine (FOWT). We design an autoencoder architecture, where the latent space represents the features of the target damaged condition. The inverse operator (encoder) is a Deep Neural Network that maps the measurable response to the parameters (means, variances, and weights) of a multivariate Gaussian Mixture model. The Gaussian Mixture model provides a convenient distributional description that is flexible enough to accommodate complex solution spaces. The decoder receives samples from the Gaussian Mixture and maps the damaged condition (states) to the system's measurable response. In such a problem, and depending on the quantities being observed (sensor positioning), it is possible that multiple damaged states may correspond to similar measurement records. In this context, the main contribution of this work lies in developing a method to quantify the uncertainty within the context of a possibly ill-posed damage identification problem. We employ the Gaussian Mixture to express the multimodal solution space and explain the uncertainty in the damaged condition estimates. We design and validate the methodology using synthetic data from a FOWT in the commonly adopted OpenFAST software and consider two damage types frequently occurring in mooring lines: biofouling and anchor displacement. The method allows for estimating the damaged state while capturing the uncertainty in the estimations and the multimodality of the solution under the availability of a limited number of response measurements.

## 1 Introduction

Floating offshore wind (FOW) is rapidly emerging as a leading form of renewable energy. Faster and steadier winds and a much larger area for future deployment are only a few appealing aspects that make FOW intriguing for industry experts and academics. While FOW's current global capacity barely exceeds 180 megawatts, recent reports predict that FOW will generate over 250 gigawatts by 2050 GWEC (2023). One of the critical barriers limiting the commercial viability of floating offshore wind turbines (FOWTs) is their operational expense, which relates to inspection, monitoring, and faulty component

replacement. These tasks are troublesome and expensive in offshore environments, whose access is compromised by weather conditions (Nava et al. (2022)).

One of the most critical components of FOWTS is the mooring system. Mooring systems, which inherit the characteristics of similar devices conceived for the oil and gas industry (Årdal et al. (2014)), employ steel chains, wire, and cable spanning dozens of meters to anchor the platforms to the seabed or to connect them. They operate under a broad range of environmental and operational conditions that may lead to damage due to corrosion, wear, and fatigue (Li et al. (2018); Liu et al. (2020)). These phenomena affect the system's behavior and may compromise the integrity of the platform if no action is taken. Ensuring the safe and optimized performance of mooring lines is crucial to minimizing operational costs and maximizing the profitability of FOWTs.

Condition monitoring of mooring systems is most often carried out on-site through visual underwater inspections (Martinez-Luengo et al. (2016)) and ultrasonic testing (Thibbotuwa et al. (2022)). Both methods are costly and often inefficient, owing to the requirement for engaging qualified manpower and appropriate equipment. Companies must also wait for acceptable weather conditions before deploying their crews on FOWT farms to comply with safety requirements. For these reasons, Structural Health Monitoring (SHM) techniques have emerged as a solution to provide a continuous, efficient, and remote assessment of these assets (Ciuriuc et al. (2022); Liu et al. (2023)). SHM typically targets an inverse problem solution that aims to identify the condition of a target system and possibly characterize associated damage based on the indirect information delivered via measurements from an instrumentation system (Farrar and Worden (2013)). Two main approaches exist in the field of SHM for FOWTs: model (or physics-based, also known as hybrid) and data-based schemes (Liang et al. (2024); Liu et al. (2022b)). On the one hand, physics-based models of dynamic systems often employ complex ordinary or partial differential equations (PDE) that govern the physical phenomena under study. While they require a deeper insight into the underlying physics, these techniques can achieve higher accuracy and generalization at the expense of computational effort (Jonkman (2007); Wang (2015); Hall and Goupee (2015)).

On the other hand, data-driven techniques employ extensive datasets to fit the desired outcome (Martinez-Luengo et al. (2016)). They can infer highly complex, nonlinear relations, provided that these are witnessed in the available data adopted for training purposes. In the context of SHM for mooring systems, indirect response sensors such as gyroscopes, inclinometers, and GPS trackers have become increasingly attractive (Gorostidi et al. (2023); Coraddu et al. (2024)). These devices are relatively inexpensive and easy to deploy while being sensitive to the presence of damage. Experimental data are noisy, constrained to practically measurable quantities, and are often limited to a specific condition (e.g., the healthy state), which covers only a subset of the inverse problem's solution subspace. Simulations from a computational parametrization (e.g., a Finite Element model) are often employed to complement experimental data and overcome this scarcity (Figueiredo et al. (2010); Zhang and Sun (2021); Fernandez-Navamuel et al. (2023)).

In the past few decades, Machine Learning (ML) algorithms have gained popularity thanks to advances in data acquisition and transmission, informatics, and computational resources. Particularly interesting are Deep Neural Networks (DNNs), which present advantageous properties such as satisfying the theorem of universal approximation (Hornik et al. (1989)) and enabling the incorporation of physical knowledge (Rojas et al. (2024)). One key challenge in the resolution of inverse problems is han-

dling uncertainties. Many authors have already dealt with uncertainty when solving inverse problems in other fields. In civil infrastructure, Betancourt *et al.* (Betancourt et al. (2021)) implemented a deep interval neural network to classify damage in a benchmark bridge. Huang *et al.* (Huang and Beck (2015)) tested the performance of a sparse Bayesian probabilistic model against incomplete data. Teimouri *et al.* (Teimouri et al. (2017)) designed a Gaussian process-based method to monitor the integrity of a composite airfoil structure. In geophysics, Alyaev *et al.* (Alyaev and Elsheikh (2022)) implemented a mixture density network for the inversion of gamma-ray logs. Liu *et al.* (Liu et al. (2022a)) estimated subsurface rock and fluid properties using deep variational autoencoders. Rodriguez *et al.* (Rodriguez et al. (2023)) extended the loss formulation stated in work (Shahriari et al. (2021)) and implemented a multimodal Variational Autoencoder (MVAE) to identify subsurface material properties. In the field of wind energy, Mylonas *et al.* (Mylonas et al. (2021)) proposed a Conditional Variational Autoencoder (CVAE) to deliver uncertainty-robust long-term fatigue predictions in a wind turbine blade based on Supervisory, Control, and Data Acquisition (SCADA) signals. Mclean *et al.* (Mclean et al. (2023)) employed Gaussian Processes (GP) to account for the uncertainty present in power curve models as the damage-sensitive features. Other works propose the use of hierarchical sparse Bayesian learning to solve the system identification problem through model updating using Gibbs sampling (Huang et al. (2017a, b)). The authors validated their strategy numerically and experimentally using a relatively simple structure with limited DOFs.

Focusing on the application of SHM to assess FOWTs, a vast amount of literature exists using DNNs. For example, Chung *et al.* (Chung et al. (2020)) fed Response Amplitude Operator (RAO) data into a DNN to detect anomalies in the cross-section of mooring lines for a tension leg platform. Their research was continued by Lee *et al.* (Lee et al. (2021)), who extended their approach to catenary lines and taut mooring systems. Janas *et al.* (Janas et al. (2021)) developed a condition-agnostic convolutional neural network (CNN) to detect anomalies caused by the loss of one line in a floating oil and gas vessel, training their model with images of its horizontal displacement history. In recent work (Sharma and Nava (2024)), Sharma *et al.* combined CNNs and Auto-Regressive (AR) models to detect biological fouling, corrosion, and anchor shift-related damage in a floating platform's mooring lines. Their study obtained AR coefficient matrices from displacements and rotations for the DeepCWind OC4 platform's surge, heave, and pitch responses fed as image inputs to the CNN.

Still, many existing works employ deterministic approaches, which suffer significant limitations when tackling real-life inverse problems, such as damage identification via monitoring data. Long-term instrumentation systems are often cheap and very limited, suffering the availability of incomplete and noisy measurements. This context makes the solution to the inverse problem highly non-unique and unstable (extremely sensitive to slight changes in the input data) (Adler and Öktem (2017)). Yet in the deterministic scope, Shahriari *et al.* (Shahriari et al. (2021)) proposed a way to define the loss function of DNNs in the measurement space rather than in the ill-posed solution space via an encoder-decoder architecture and a two-step training phase. With this strategy, they constrained the training and prevented undesired solutions. Based on this idea, Gorostidi *et al.* (Gorostidi et al. (2023)) attempted to detect failures in the mooring system of a FOWT based on response measurements. They employed statistics-based features from six Degrees Of Freedom (DOFs) to identify the level of biofouling and anchoring damage, using synthetic data from a FOWT simulated in Openfast (Jonkman et al. (2022)). Although this approach aids in identifying a physically plausible scenario, it neglects the multimodality of the solution (i.e., various damage scenarios producing the same

measured response). Addressing this ill-posedness requires moving from deterministic to probabilistic approaches to quantify the uncertainty in the solution space and provide more reliable assessments.

However, there is still a considerable gap in implementing uncertainty-aware methods for the condition assessment of mooring systems in FOWTs. This work intends to contribute to this direction. Here, we extend our previous work (Gorostidi et al. (2023, 2022)) that aimed at inferring the underlying health condition of mooring systems using response measurements, which delivered a deterministic assessment. Following the recent work by Rodriguez *et al.* (Rodriguez et al. (2023)), we propose a Bayesian approach for quantifying the uncertainty in the delivered damage estimates. Our proposal adapts the multimodal VAE methodology introduced in (Goh et al. (2021); Rodriguez et al. (2023)) to solve the inverse problem of damage identification of FOWT mooring systems. The core of the proposed methodology is to probabilistically describe the solution to an inverse problem that may exhibit high multimodality. This situation frequently occurs when dealing with sparse instrumentation systems, which is often the case in SHM applications (Teughels and De Roeck (2004); Oliveira et al. (2018a, b); Devriendt et al. (2014); Adão da Fonseca and Bastos (2004)).

We design an encoder-decoder architecture to address ill-posed inverse problems within the context of damage identification. According to the forward operator, the proposed methodology ensures that the estimated solutions are physically meaningful. We describe the multimodal solution space (damaged condition) using a multivariate Gaussian Mixture. This parametrized distributional model is mathematically convenient to integrate into the differentiable scheme of DNNs and is sufficiently flexible to accommodate complex distributions. The inverse operator (encoder) estimates the parameters (i.e., means, variances, and weights) that build a corresponding Gaussian Mixture describing the distribution of the damage condition features. We draw samples from the estimated posterior distribution model that are subsequently fed into the forward approximation (decoder). With this strategy, we statistically describe the target damaged condition space, accounting for the multimodality of the solution, which occurs mainly when the instrumentation system comprises a limited amount of sensors.

Training the inverse produces the posterior distribution that describes the damaged condition given some input measurements of the system's response. In this work, statistical features of measured rotational DOFs are employed as the response measurements. We employ rotations since these can be measured experimentally through low-cost, low-maintenance sensors, similar to acceleration, but still delivering information that reflects the resulting response. A benefit of using rotations lies in delivering quantities that contain lower frequency information with respect to acceleration signals. This is useful when the involved response includes lower-frequency components (e.g., rigid body motions and drifts). The loss function leverages two terms; the first accounts for the measurement misfit, while the second drives the shape of the posterior distribution.

We analyze the effect of uncertainty in the delivered estimates of our proposed scheme in eight test examples that correspond to representative damage scenarios. We further explore how using one single DOF increases the ill-posedness of the inverse SHM problem, demonstrating the strength of our proposed method in providing a more reliable diagnosis with poor instrumentation systems. Finally, we further investigate the robustness of the methodology to different measurement error levels. Results demonstrate that our method successfully captures the uncertainty in the predictions, describing the multimodality of the solution mainly in the absence of some response signals.

Despite the successful results, this work suffers certain limitations that ought to be acknowledged. First, the methodology provides a way to describe the uncertainty that is inherent to the ill-posedness of the inverse problem. Such uncertainty induces the multimodality of the output (i.e., the damage condition estimate). The method also reflects the effect of aleatory uncertainty (noisy measurements) as it transfers from the measured data to the estimated condition. However, this work neglects the epistemic uncertainty, which occurs when making a prediction on measurements that correspond to a damage condition that

is far from those employed in the training stage. Accounting for such uncertainty and disentangling both sources is beyond the scope of this work and requires further study. Second, since the Gaussian mixture is a parametric approach, it constrains the outcome and prevents a complete characterization of the effect of uncertainty. Improving the distributional model requires many components, which enormously increases the number of parameters to be estimated. This is an important limitation of the method, mostly when scaling up to higher-dimensional spaces. Finally, due to the current scarcity of experimental data

from real operating FOWTs, this work is entirely restricted to synthetic data from computational simulations. Integrating experimental data with synthetic scenarios is a key challenge to proving the applicability of the suggested methodology in real-field data.

    The remainder of this article is structured as follows. Section 2 derives the mathematical formulation describing the inverse problem. Section 3 presents the turbine-platform assembly and the excitation and damaged conditions employed in our simu-

140 lations. Section 4 describes the specifications of the proposed DNN architecture and training stage. Sections 5 and 6 discuss the method's performance considering (i) three DOFs and (ii) a single DOF. Finally, Section 7 highlights the conclusions and limitations of the proposed work and reveals future research lines.

## 2   Methodology

We describe the rigid body response of a FOWT platform in terms of six degrees of freedom (DOFs): surge (forward-backward

motion), sway (sideways motion), heave (vertical motion), roll (rotation about the longitudinal axis), pitch (rotation about the transverse axis), and yaw (rotation about the vertical axis) (Tran and Kim (2015)). Since transmission of time-domain signals is extremely expensive, their content is often condensed in the form of statistical features, which is the typical approach to storing SCADA data (Gorostidi et al. (2023)). Let $\mathbf{m} \in \mathcal{M}$ denote the platform's response with $M$ features extracted from the time-domain signals of the six DOFs. This response relates to the loading and the system properties through a set of partial

differential equations (PDEs). These PDEs describe the aerodynamics, hydrodynamics, servodynamics, and elastodynamics of the coupled system operating under wind and wave excitation conditions $\mathbf{w} \in \mathcal{W}$ (Tran and Kim (2015)).

    Damage in the mooring system affects its physical properties (e.g., stiffness). We denote by $\mathbf{z} \in \mathcal{Z}$ the set of damage features describing the condition of the FOWT mooring system. These features indicate the level of each possible existing damage and succinctly represent the changes in the coefficients of the governing PDEs. We define $\mathcal{F} : \mathcal{Z} \times \mathcal{W} \to \mathcal{M}$ as the forward operator

mapping the damage and loading conditions to the response of the system. Here, $\mathcal{F}$ includes the PDEs governing the response of the floating platform of the FOWT. The cartesian product of the domains $\mathcal{Z}$ and $\mathcal{W}$, represents all combinations of damage and loading conditions.

## 2.1 Deterministic inverse operator

In damage identification, we seek for the inverse operator $\mathcal{I} : \mathcal{M} \times \mathcal{W} \to \mathcal{Z}$ which, given some noisy response measurements $\mathbf{m}$ under a prescribed excitation $\mathbf{w}$, yields the system's damaged condition $\mathbf{z}$. This relationship is often unknown and highly nonlinear. Here, we approximate $\mathcal{I}$ using Deep Neural Networks (DNNs). DNNs have remarkable power in approximating complex nonlinear functions (Hornik (1991)). Moreover, once trained, such models produce evaluations in milliseconds. These particularities render DNNs appealing in many areas, including damage identification. Let $\mathcal{I}_{\boldsymbol{\theta}}$ be the DNN described by parameters $\boldsymbol{\theta}$ that approximates the inverse problem $\mathcal{I}$. For a certain observed input $[\mathbf{m}, \mathbf{w}]$, we define a loss function to measure the discrepancy between the estimates of the approximate inverse $\mathcal{I}_{\boldsymbol{\theta}}$ and the true damaged condition of the system, $\mathbf{z}$:

$$\mathcal{L}_{\mathcal{Z}}(\boldsymbol{\theta}) = \|\mathbf{z} - \mathcal{I}_{\boldsymbol{\theta}}([\mathbf{m}, \mathbf{w}])\|_2^2, \tag{1}$$

where we employ the squared $l_2$ norm as the discrepancy metric. We can find the optimal parameter set $\boldsymbol{\theta}$ by minimizing the loss function in Eq. 1 over a training dataset that contains $D$ labeled responses $\mathcal{D} = \{\mathbf{m}_i, \mathbf{w}_i, \mathbf{z}_i\}_{i=1}^D$.

However, in real physical problems, the instrumentation system is often sparse and thus not fully sensitive to the targeted damage scenarios. This situation renders the forward operator $\mathcal{F}$ non-injective, i.e., more than one system state (damaged condition) may produce the same response (Tarantola (2004)). Consequently, the inverse problem is ill-posed, meaning it might have multiple solutions for the same noisy input. Given this ill-posedness of the inverse problem under incomplete and noisy data availability, minimizing $\mathcal{L}_{\mathcal{Z}}(\boldsymbol{\theta})$ may produce an infeasible outcome that simply results by averaging possible candidate solutions. Thus, defining the loss on the space of the damaged conditions $\mathcal{Z}$ is inconvenient. To overcome this issue, we adopt an encoder-decoder strategy, where the encoder approximates the inverse problem (given the measurements $\mathbf{m}$, estimate the damage condition features $\mathbf{z}$), and the decoder corresponds to the forward operator. With this architecture, the damage condition estimates are enforced to satisfy the forward operator (governing ordinary or partial differential equations) and thus are consistent with the underlying physics describing the system's behavior. This composition of the forward with the inverse, which constitutes the identity mapping, enables expressing the loss function on the space of the measured responses, $\mathcal{M}$:

$$\mathcal{L}_{\mathcal{M}}(\boldsymbol{\theta}) = \|\mathbf{m} - \mathcal{F} \circ \mathcal{I}_{\boldsymbol{\theta}}([\mathbf{m}, \mathbf{w}])\|_2^2, \tag{2}$$

using the $l_2$ norm. Minimization of this loss function ensures that the DNN will report one out of all feasible solutions.

The main bottleneck when minimizing the loss function in Eq. 2 owes to the effort required to massively evaluate the forward operator $\mathcal{F}$. We define a DNN $\mathcal{F}_{\phi}$ described by parameters $\phi$ that approximates the forward operator $\mathcal{F}$. Once trained, $\mathcal{F}_{\phi}$ provides computationally efficient system responses that substitute the expensive forward evaluations. Figure 1 graphically describes the connection of the forward with the inverse operator using a fully connected architecture.

We adopt a two-step training strategy proposed and employed in previous works (Shahriari et al. (2021); Rodriguez et al. (2023); Gorostidi et al. (2023)). The contribution of these works and their connection with the present approach is extensively

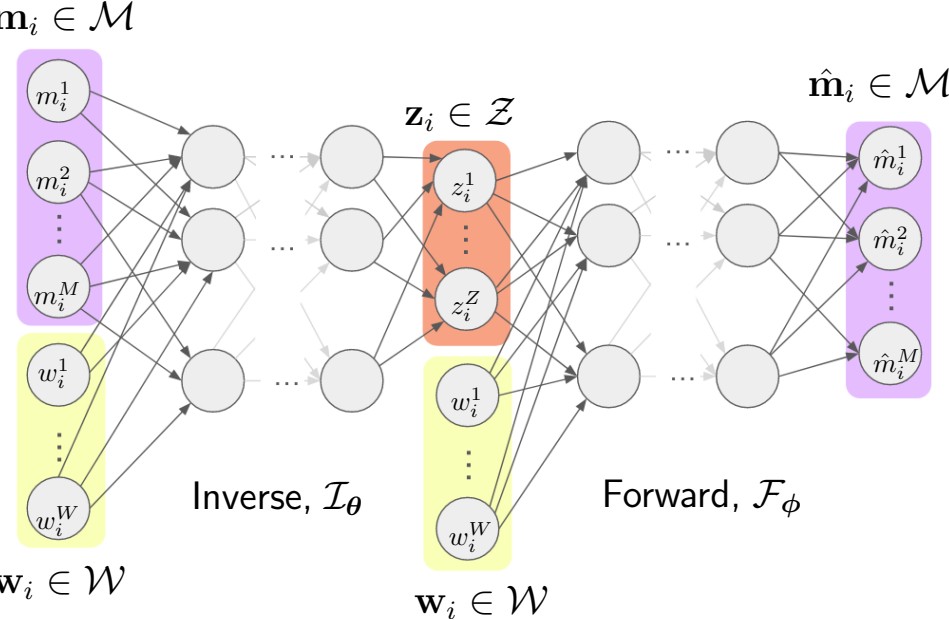

**Figure 1.** DNN architecture connnecting the forward $\mathcal{F}_\phi$ with the inverse $\mathcal{I}_\theta$. The output of the entire DNN yields the reconstruction of the input measurements, $\hat{\mathbf{m}}_i = \mathcal{F}_\phi \circ \mathcal{I}_\theta(\mathbf{m}_i)$. This architecture permits defining the loss function on the measurements space, $\mathcal{M}$. The wind and wave excitations $\mathbf{w}$ are fed to both the forward and the inverse. For representation feasibility, we depict the distribution as a bi-variate Gaussian mixture

described in section 1. We first obtain the optimal forward parameters $\phi^*$:

$$\phi^* := \arg\min_{\phi} \|\mathbf{m} - \mathcal{F}_\phi([\mathbf{z}, \mathbf{w}])\|_2^2. \tag{3}$$

Then, we find the optimal inverse parameters $\theta^*$:

$$\theta^* := \arg\min_{\theta} \|\mathbf{m} - (\mathcal{F}_\phi^* \circ \mathcal{I}_\theta)([\mathbf{m}, \mathbf{w}])\|_2^2. \tag{4}$$

The second step incorporates $\mathcal{F}_\phi^*$ as a non-trainable architecture.

After the two-step training, we essentially are interested in using $\mathcal{I}_{\theta^*}$ to estimate the damaged condition $\mathbf{z}$ of the FOWT from the measured responses $\mathbf{m}$ and operational conditions $\mathbf{w}$. In this deterministic framework, the inverse operator outputs a single-value estimate for each damaged condition given an input $[\mathbf{m}, \mathbf{w}]$. This approach precludes adequate interpretation of the uncertainty in the estimates, which are assumed to be $100\%$ confident. However, despite ensuring that the provided estimate corresponds to one of the possible solutions (assuming an adequate and successful training process), this might differ from the actual condition of the system.

## 2.2 Bayesian inverse operator

In this work, we overcome the limitation of the deterministic solver by adopting a Bayesian approach to approximate the inverse operator $\mathcal{I}$ (Rodriguez et al. (2023); Goh et al. (2021)). Since the physical law governing the system is deterministic, we consider the forward as a deterministic operator. We assume that uncertainty resulting from recording noisy measurements propagates from the data to the predicted damage condition estimates (through the inverse operator). Hence, instead of mapping the measurements to the damaged condition, we seek the operator that maps the input probability distribution (of the measurements) to the distribution of the damaged condition. The distribution of the damaged state features is conditioned on the measured data, i.e., we search for a conditional probability distribution.

Let us assume that the measured response, loading conditions, and unknown system's damaged conditions are represented as random variables. Vectors $\mathbf{m}$, $\mathbf{w}$, and $\mathbf{z}$, correspond to realizations of each of these sets of random variables. Considering the existence of additive noise that is inherent to the data acquisition process, we express the forward problem as:

$$\mathbf{m} = \mathcal{F}(\mathbf{z}, \mathbf{w}) + \boldsymbol{\epsilon}, \tag{5}$$

where we substitute $\mathcal{F}$ by its optimal approximation $\mathcal{F}_{\phi^*}$, and $\boldsymbol{\epsilon}$ is the unknown measurement error with known statistics described by a Probability Density Function (PDF) $\boldsymbol{\epsilon} \sim p(\boldsymbol{\epsilon})$. Although $\mathbf{z}$ is unknown, we can represent its uncertainty using a conditional probability distribution considering its relationship with the measured variables, $p(\mathbf{z}|\mathbf{m}, \mathbf{w})$. For any damaged condition $\mathbf{z}$, the conditional PDF $p(\mathbf{z}|\mathbf{m}, \mathbf{w})$ is the target posterior distribution that forms the estimated target of the inverse problem $\mathcal{I}$ under the use of a Bayesian approach. We employ Bayes' theorem to express it by the following proportionality:

$$p(\mathbf{z}|\mathbf{m}, \mathbf{w}) \propto p(\mathbf{m}|\mathbf{z}, \mathbf{w}) \cdot p(\mathbf{z}), \tag{6}$$

where $p(\mathbf{z})$ represents the prior PDF of the unknown damaged condition $\mathbf{z}$, and $p(\mathbf{m}|\mathbf{z}, \mathbf{w})$ is the likelihood model that expresses the interrelation between the measurements and the damaged condition.

Due to the intractability of the true posterior (usually an unknown non-parametric PDF), we define an inverse operator $\mathcal{I}_{\boldsymbol{\theta}}$ that estimates the parameters of an approximate PDF for each damaged condition in $\mathbf{z}$, given the response measurements $\mathbf{m}$ and operating loads $\mathbf{w}$. Since we have no prior knowledge, we approximate this PDF by a flexible one, given by a mixture of multivariate Gaussian functions (Deisenroth et al. (2020)), such that:

$$q_{\boldsymbol{\theta}}(\mathbf{z}|\mathbf{m}, \mathbf{w}) \sim GMM(\boldsymbol{\mu}, \boldsymbol{\sigma}) = \sum_{k=1}^{K} \alpha_k \mathcal{N}(\mathbf{z}|\boldsymbol{\mu}_k, \Sigma_k);$$
$$\mathcal{N}(\mathbf{z}|\boldsymbol{\mu}_k, \Sigma_k) = \frac{1}{(2\pi)^{Z/2}|\Sigma_k|^{1/2}} exp\left(-\frac{1}{2}(\mathbf{z} - \boldsymbol{\mu}_k)\Sigma_k^{-1}(\mathbf{z} - \boldsymbol{\mu}_k)\right), \tag{7}$$

where the PDF $q_{\boldsymbol{\theta}}(\mathbf{z}|\mathbf{m}, \mathbf{w})$ is a Gaussian Mixture Model (GMM) dependent on the parameters $\boldsymbol{\theta}$ of the DNN $\mathcal{I}_{\boldsymbol{\theta}}$, $\alpha_k$ is the weight of the $k$-th Gaussian in the mixture, and $\mathcal{N}(\mathbf{z}|\boldsymbol{\mu}_k, \Sigma_k)$ is the corresponding multivariate Gaussian distribution with mean vector $\boldsymbol{\mu}_k$ and diagonal covariance matrix $\Sigma_k = diag(\boldsymbol{\sigma}_k)$ for $k = 1, ..., K$ Gaussians. Compared to the deterministic approach, instead of a single value estimate, now the output of the inverse operator $\mathcal{I}_{\boldsymbol{\theta}}$ spans to produce the set of properties describing

the mixture, namely the vector $GMM_{props} = \{\boldsymbol{\mu}_1, ..., \boldsymbol{\mu}_K, \boldsymbol{\sigma}_1, ....\boldsymbol{\sigma}_K, \alpha_1, ..., \alpha_K\}$. This results in an output of $K(2 \times Z + 1)$ dimensions. For simplicity in notation, we have omitted the dependency of the GMM properties on the DNN parameters $\boldsymbol{\theta}$.

Realizations from $q_{\boldsymbol{\theta}}(\mathbf{z}|\mathbf{m}, \mathbf{w})$ represent samples of the damaged condition $\mathbf{z}$ that are likely produced by the unknown true posterior $p(\mathbf{z}|\mathbf{m}, \mathbf{w})$. All the realizations $\{\mathbf{z}_h\}_{h=1}^{H}$ share the same operating conditions $\mathbf{w}$. Recovering the strategy of composing the forward with the inverse (see Figure 1), we feed the samples to the optimal forward operator $\mathcal{F}_{\phi^*}$ trained in Eq. 3. Figure 2 schematically represents the architecture in the Bayesian approach.

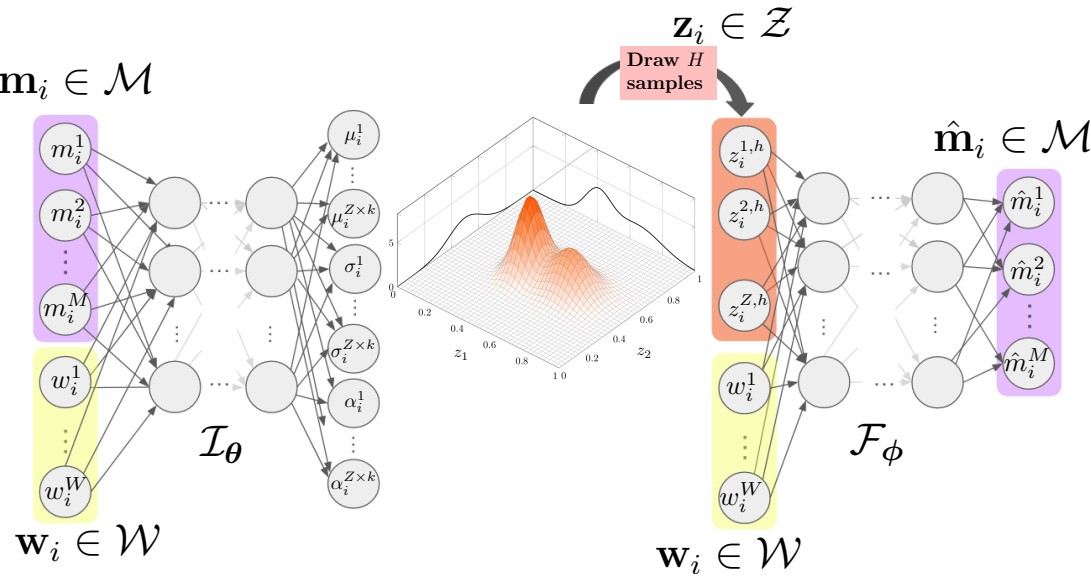

**Figure 2.** DNN architecture in the adopted Bayesian approach. The output of the inverse operator produces the parameters that describe the PDF of the damaged condition $\mathbf{z}$. A sampling layer draws $H$ random samples from the distribution, which are then fed to the optimal forward operator $\mathcal{F}_{\phi^*}$. The output of the entire DNN yields the reconstruction of the input measurements, $\hat{\mathbf{m}}_i = \mathcal{F}_{\phi} \circ \mathcal{I}_{\boldsymbol{\theta}}(\mathbf{m}_i)$.

We aim at minimizing the discrepancy between the true posterior $p(\mathbf{z}|\mathbf{m}, \mathbf{w})$ and the approximate posterior $q_{\boldsymbol{\theta}}(\mathbf{z}|\mathbf{m}, \mathbf{w})$ obtained from $\mathcal{I}_{\boldsymbol{\theta}}$. We first assume that noise $\boldsymbol{\epsilon}$ is mutually independent with respect to the unknown damaged condition $\mathbf{z}$ (Goh et al. (2021)). Thus, according to Eq. 5, we can express the likelihood as:

$$p(\mathbf{m}|\mathbf{z}, \mathbf{w}) = p(\mathbf{m} - \mathcal{F}([\mathbf{z}, \mathbf{w}])), \tag{8}$$

We then assume the noise follows a Gaussian distribution, $p(\boldsymbol{\epsilon}) = \mathcal{N}(0, \Gamma)$, where $\Gamma = diag(\beta \mathcal{F}([\mathbf{z}, \mathbf{w}]))^2$ is a vector that contains the non-zero elements of a diagonal matrix, and the parameter $\beta$ corresponds to the noise level. This allows us to rewrite Eq. 6 as:

$$p(\mathbf{z}|\mathbf{m}, \mathbf{w}) \propto p(\mathbf{m} - \mathcal{F}(\mathbf{z}, \mathbf{w})) \cdot p(\mathbf{z}) = \frac{1}{(2\pi)^{M/2}|\Gamma|^{1/2}} exp\left(-\frac{1}{2}(\mathbf{m} - \mathcal{F}([\mathbf{z}, \mathbf{w}]))^t \Gamma^{-1}(\mathbf{m} - \mathcal{F}([\mathbf{z}, \mathbf{w}]))\right), \tag{9}$$

where $p(\mathbf{z})$ is the prior distribution of the damaged condition properties, which can follow any PDF. The Kullback-Leibler divergence (KL) is defined as a statistical measure of the distance between two PDFs (Deisenroth et al. (2020); Asperti and Trentin (2020)). KL assumes a null value when the compared PDFs are equal and can be defined as:

$$KL[p(x)||q(x)] = \int p(x)\, log \frac{p(x)}{q(x)} dx, \tag{10}$$

where x indicates a realization of the random variable $X$, and $p(x)$ and $q(x)$ are the two different PDFs. Here, we use the KL metric to evaluate the distance between the true posterior $p(\mathbf{z}|\mathbf{m},\mathbf{w})$ and its approximation estimated by the inverse operator $\mathcal{I}_{\boldsymbol{\theta}}$, denoted as $q_{\boldsymbol{\theta}}(\mathbf{z}|\mathbf{m},\mathbf{w})$, yielding:

$$KL[q_{\boldsymbol{\theta}}(\mathbf{z}|\mathbf{m},\mathbf{w})||p(\mathbf{z}|\mathbf{m},\mathbf{w})] = \int q_{\boldsymbol{\theta}}(\mathbf{z}|\mathbf{m},\mathbf{w})\, log \frac{q_{\boldsymbol{\theta}}(\mathbf{z}|\mathbf{m},\mathbf{w})}{p(\mathbf{z}|\mathbf{m},\mathbf{w})} dz. \tag{11}$$

However, the KL divergence term exhibits certain shortcomings that weaken its strength as a distance metric; it is asymmetric, it does not satisfy the triangle inequality, and it produces an intractable term (the evidence of the data distribution $p(\mathbf{m})$) (Blei et al. (2017)). Instead, a lower bound is calculated for the evidence, known as the Evidence Lower BOund (ELBO) (Blei et al. (2017)). ELBO is the loss function commonly employed in Variational Autoencoders (VAEs) to account for the discrepancy between the two distributions (Goh et al. (2021); Rodriguez et al. (2023)). We obtain the ELBO loss by exploiting the KL expression and isolating the intractable terms::

$$
\begin{aligned}
KL[q_{\boldsymbol{\theta}}(\mathbf{z}|\mathbf{m},\mathbf{w})||p(\mathbf{z}|\mathbf{m},\mathbf{w})] &= \int q_{\boldsymbol{\theta}}(\mathbf{z}|\mathbf{m},\mathbf{w}) \log \frac{q_{\boldsymbol{\theta}}(\mathbf{z}|\mathbf{m},\mathbf{w})}{p(\mathbf{z}|\mathbf{m},\mathbf{w})}\, dz \\
&= \int q_{\boldsymbol{\theta}}(\mathbf{z}|\mathbf{m},\mathbf{w}) \log q_{\boldsymbol{\theta}}(\mathbf{z}|\mathbf{m},\mathbf{w})\, dz - \int q_{\boldsymbol{\theta}}(\mathbf{z}|\mathbf{m},\mathbf{w}) \log p(\mathbf{z}|\mathbf{m},\mathbf{w})\, dz \\
&= \mathbb{E}_q[\log q_{\boldsymbol{\theta}}(\mathbf{z}|\mathbf{m},\mathbf{w})] - \mathbb{E}_q[\log p(\mathbf{z}|\mathbf{m},\mathbf{w})] = \\
&= \mathbb{E}_q[\log q_{\boldsymbol{\theta}}(\mathbf{z}|\mathbf{m},\mathbf{w})] - \mathbb{E}_q[\log \frac{p(\mathbf{m},\mathbf{w},\mathbf{z})}{p(\mathbf{m},\mathbf{w})}] \\
&= \mathbb{E}_q[\log q_{\boldsymbol{\theta}}(\mathbf{z}|\mathbf{m},\mathbf{w}) - \log p(\mathbf{m},\mathbf{w},\mathbf{z})] + \log p(\mathbf{m},\mathbf{w}).
\end{aligned} \tag{12}
$$

In Eq. 12, we employ the relationship between the joint distribution and the posterior: $p(\mathbf{z}|\mathbf{m},\mathbf{w}) = \frac{p(\mathbf{m},\mathbf{w},\mathbf{z})}{p(\mathbf{m},\mathbf{w})}$. We rearrange the terms and define the ELBO as:

$$
\begin{aligned}
ELBO = \log p(\mathbf{m},\mathbf{w}) - KL[q_{\boldsymbol{\theta}}(\mathbf{z}|\mathbf{m},\mathbf{w})||p(\mathbf{z}|\mathbf{m},\mathbf{w})] &= \mathbb{E}_q[\log p(\mathbf{m},\mathbf{w},\mathbf{z}) - \log q_{\boldsymbol{\theta}}(\mathbf{z}|\mathbf{m},\mathbf{w})] \\
&= \mathbb{E}_q[\log(p(\mathbf{m}|\mathbf{w},\mathbf{z})p(\mathbf{w})p(\mathbf{z}))] - \mathbb{E}_q[\log q_{\boldsymbol{\theta}}(\mathbf{z}|\mathbf{m},\mathbf{w})] \\
&= \mathbb{E}_q[\log p(\mathbf{m},\mathbf{w}|\mathbf{z})] + \mathbb{E}_q[\log p(\mathbf{z})] - \mathbb{E}_q[\log q_{\boldsymbol{\theta}}(\mathbf{z}|\mathbf{m},\mathbf{w})],
\end{aligned} \tag{13}
$$

where we assume that the operating conditions $\mathbf{w}$ and the damage properties $\mathbf{z}$ are independent (i.e., $p(\mathbf{w},\mathbf{z}) = p(\mathbf{w}) \cdot p(\mathbf{z})$). Hence, we can remove $p(\mathbf{w})$ in the second line of the equation, as it is independent of $\mathbf{z}$.

For a certain observation $\{\mathbf{m},\mathbf{w}\}$, we draw $H$ samples from the posterior and approximate the ELBO loss function as

$$\mathcal{L}_{ELBO}(\boldsymbol{\theta}) \approx \frac{1}{H} \sum_{h=1}^{H} [\underbrace{\log p(\mathbf{m},\mathbf{w}|\mathbf{z}^h)}_{\text{Likelihood}} + \underbrace{\log p(\mathbf{z}^{\mathbf{h}})}_{\text{Prior}} - \underbrace{\log q_{\boldsymbol{\theta}}(\mathbf{z}^h|\mathbf{m},\mathbf{w})}_{\text{Approx. posterior}}]. \tag{14}$$

By substituting the likelihood from Eq. 9 into Eq. 14, we finally express $\mathcal{L}_{ELBO}$ as:

$$\mathcal{L}_{ELBO}(\boldsymbol{\theta}) \approx \frac{1}{H}\sum_{h=1}^{H}[-\frac{1}{2}(\mathbf{m}-\mathcal{F}([\mathbf{z}^h,\mathbf{w}]))^t\Gamma^{-1}(\mathbf{m}+\mathcal{F}([\mathbf{z}^h,\mathbf{w}]))+\log p(\mathbf{z}^h)-\log q_{\boldsymbol{\theta}}(\mathbf{z}^h|\mathbf{m},\mathbf{w})]. \tag{15}$$

Here, the first term accounts for the data misfit, which is the error between the true measurements and the reconstructions provided by $\mathcal{F}_{\phi^*}$. The second term refers to the prior, which we assume to follow a bounded uniform distribution $p(\mathbf{z}) \sim \mathcal{U}[\mathbf{b}_{low},\mathbf{b}_{up}]$ with lower and upper bounds $\mathbf{b}_{low}$ and $\mathbf{b}_{up}$, respectively. The last term measures the probability that the $h$-
th sample belongs to the estimated distribution $q_{\boldsymbol{\theta}}(\mathbf{z}|\mathbf{m},\mathbf{w})$. The second term can be neglected by directly constraining the Gaussian mixture density function to the desired interval (according to the assumption of uniform distribution). Note that minimizing the Kullback Leibler divergence between the original and the estimated posteriors ($KL[q_{\boldsymbol{\theta}}(\mathbf{z}|\mathbf{m},\mathbf{w})||p(\mathbf{z}|\mathbf{m},\mathbf{w})]$) is equivalent to maximizing $\mathcal{L}_{ELBO}$ or minimizing its negative. For a training dataset $\mathcal{D}$ with $D$ labeled observations ($\mathcal{D} = \{\mathbf{m}_i,\mathbf{w}_i,\mathbf{z}_i\}_{i=1}^{D}$), and $H$ drawn samples from the estimated PDF, we obtain the optimal parameter set $\boldsymbol{\theta}^*$ by minimizing the
negative of $\mathcal{L}_{ELBO}$:

$$\boldsymbol{\theta}^* := \underset{\boldsymbol{\theta}}{\arg\min}\ \frac{1}{D\cdot H}\sum_{i=1}^{D}\sum_{h=1}^{H}\left[\frac{1}{2}(\mathbf{m}_i-\mathcal{F}([\mathbf{z}_i^h,\mathbf{w}_i]))^t\Gamma^{-1}(\mathbf{m}_i-\mathcal{F}([\mathbf{z}_i^h,\mathbf{w}_i]))+log\ q_{\boldsymbol{\theta}}(\mathbf{z}_i^h|\mathbf{m}_i,\mathbf{w}_i)\right]. \tag{16}$$

## 3   Case study

We consider the use case of the 5MW FOWT designed by the National Renewable Energy Laboratory (NREL) (Jonkman et al. (2009)) mounted atop the DeepCWind semi-submersible platform (Robertson et al. (2014a)). The model was developed as part
of the Offshore Code Comparison Collaboration Continuation (OC4) project, which aimed to verify the accuracy of offshore wind turbine dynamics simulation tools through extensive code-to-code comparisons. This involved contributions from numerous organizations worldwide, ensuring a robust and well-validated design. The semisubmersible floater for OC4 was specifically designed to serve as a benchmark for offshore wind energy research and development purposes (Robertson et al. (2014b)). The motivation for adopting such a turbine-floater assembly in this study lies in its computational availability and popularity
within the FOW research community. Moreover, the OC4 model underwent experimental validation. The validation process included comparisons between numerical simulations and experimental data from wave tank tests (Borisade et al. (2018)). The OC4 floater, depicted in Figure 3, provides stability to the turbine facing unsteady and unpredictable sea conditions thanks to three partially ballasted base cylindrical columns. A set of pontoons and cross braces connects each column to the others and to the main tower. The platform is held in place by three catenary mooring lines located 120 from one another. The selected
measurements describing the platform's response include features from the three rotational DOFs: roll, pitch, and yaw. From a sensory perspective, these measurements can be experimentally obtained using inclinometers, cheap and reliable devices often used in long-term monitoring, currently also effectuated via Micro-Electro-Mechanical System (MEMS) technologies. Given the mechanical symmetry of the system due to the geometrical properties as well as the force and load distributions. Compared to acceleration signals, which are often measured in the field of FOWT condition assessment, displacements (and coupled
rotation DOFs) tend to be more sensitive under low-frequency dynamics, such as floating platform responses. In the recent

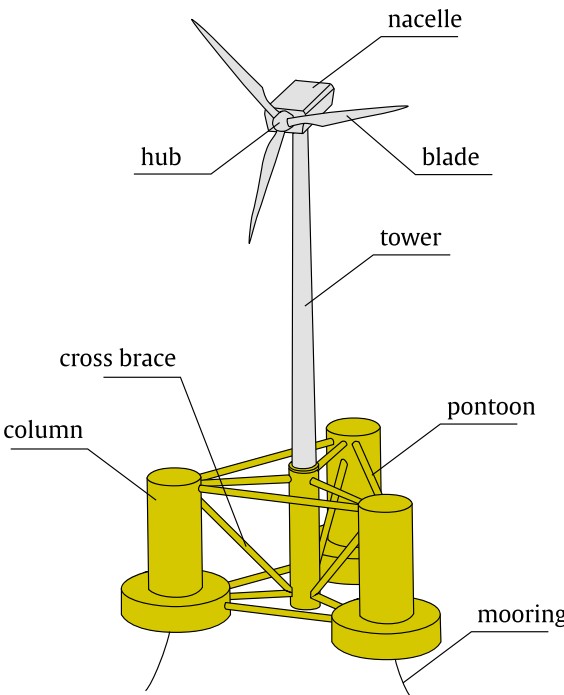

**Figure 3.** 5MW NREL FOWT mounted atop the DeepCWind OC4 semi-submersible platform.

work by Sharma et al.(Sharma and Nava (2024)), the authors analyzed the sensitivity of acceleration and displacement signals under mooring system damage (anchoring), highlighting the potential of displacements over accelerations even for low-level damage.

Figure 4 describes the response under common environmental conditions. We extract a set of modal statistics from these responses, which we assume are sufficiently descriptive of the platform's movement.

As observed in Figure 4, owing to the system's symmetry, rotations in roll DOF are harmonic oscillations around the mean position with an energy content associated with the surge and roll natural periods, while pitch exhibits a more colorful PSD being affected by the external excitation.

From the time-domain responses (see Figures 4(a), 4(c), and 4(e)), we compute the mean displacement as:

$$\bar{x} = \int_{t_0}^{t_f} x \, dt \approx \frac{1}{N} \sum_{i=1}^{N} x_i, \tag{17}$$

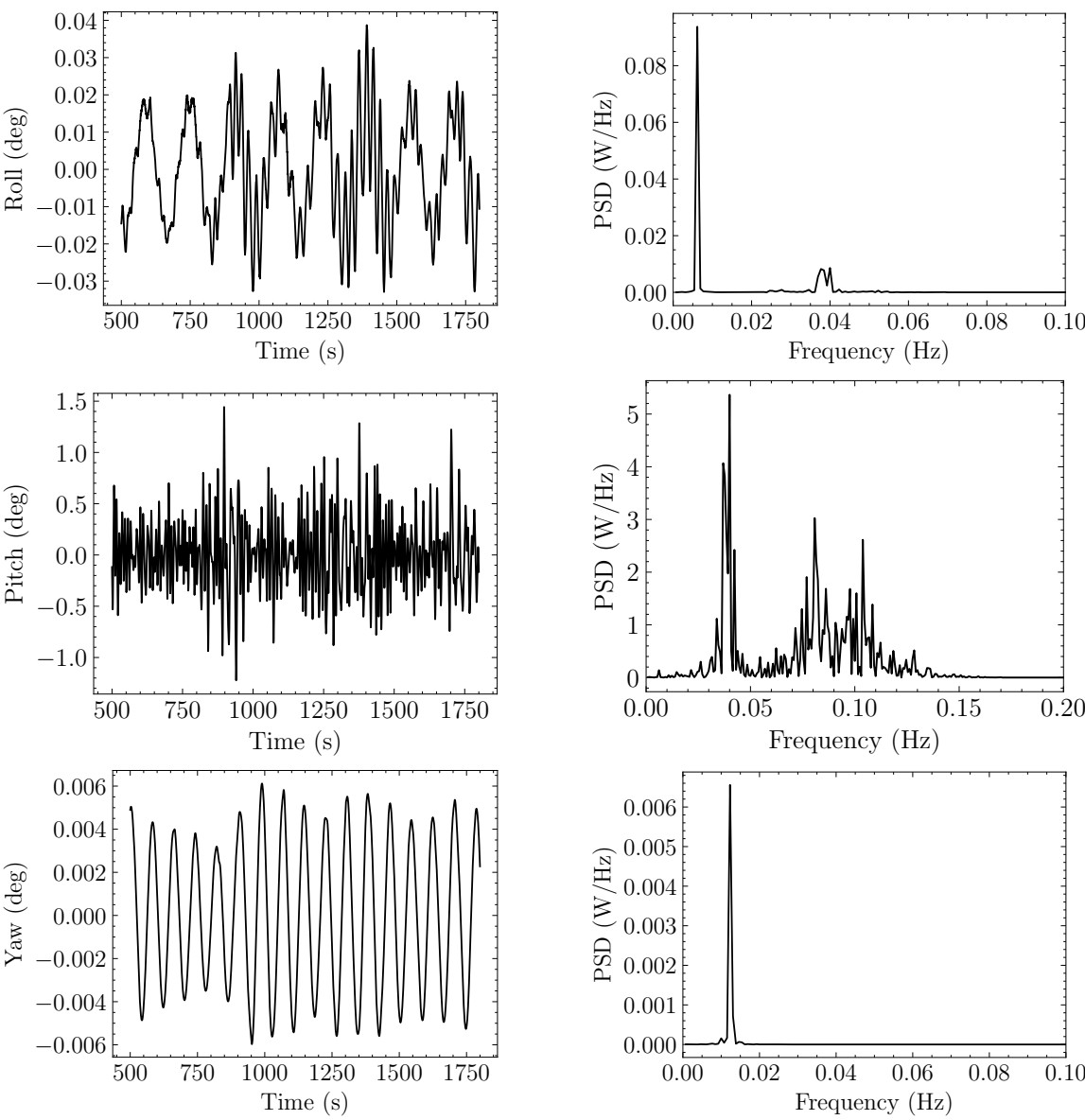

**Figure 4.** Response amplitude operator (RAO) of the FOWT for roll response in (a) time, and (b) frequency domain; for pitch response in (c) time, and (d) frequency domain; and for yaw response in (e) time, and (f) frequency domain.

where $N$ indicates the total number of data points. We further compute the standard deviation of the response as

$$\sigma = \sqrt{\frac{1}{N-1}\sum_{i=1}^{N}(x_i - \bar{x})^2}. \tag{18}$$

We assume the time domain response to be stationary by neglecting its transient state. To obtain the frequency spectra (see Figures 4(b), 4(d), and 4(f)), we compute the Power Spectral Density (PSD) of the signals, which describes the power distribution across a frequency range as (Theodoridis (2020)):

$$S_x(f) = \lim_{T \to \infty} \frac{\mathbb{E}\left[|F_x(f)|^2\right]}{2T}, \tag{19}$$

where $F_x(f)$ is the Fourier Transform of the time-domain signal for any DOF $x$. Finally, we identify two dominant peak frequencies as

$$f_1 = \underset{f \in [0, f_{thresh}]}{\arg\max} \; S_x(f), \tag{20}$$

$$f_2 = \underset{f \in [f_{thresh}, \infty]}{\arg\max} \; S_x(f). \tag{21}$$

One of the peaks for each DOF usually matches the platform's natural frequency. In contrast, the further spectral peaks reflect the influence of external conditions, e.g., wind, current, and waves, in the system's response (Benitz et al. (2014)). We have split each DOF's natural and excitation frequencies using threshold frequencies $f_{thresh}$ to ensure both peaks are identified.

To assess the magnitude of the peaks and the intensity of all the frequencies in the spectra, we also measure the zero-th momentum as (Sundar (2017)):

$$m_0 = \int\limits_0^{\omega_f} \omega S_x d\omega, \tag{22}$$

where $\omega$ is the angular frequency in radians per second. These are the five features we employ to describe the time-domain response of the platform. These statistics are ultimately the inputs of our neural network. This method allows for a significant reduction in the number of inputs required to assess the health status of mooring lines' integrity while maintaining high physical accuracy.

In this work, we use NREL's open-source wind turbine simulation tool OpenFAST (OpenFAST Documentation (Year of Access)), which evaluates the influence of aerodynamic (Jonkman et al. (2015); Platt et al. (2016)) and hydrodynamic (Jonkman et al. (2014)) excitations on the response of the floating platform. The number of deployed real-scale FOW turbines is limited; moreover, the data related to the performances of these devices are proprietary, and even if they were publicly available, the amount of labeled data under damaged conditions in the mooring systems might be extremely reduced, given the short life of these platforms. The current lack of operational data from such devices demands solutions that may exploit simulation tools in the delivery of predictive models. OpenFAST is regarded as a highly accurate and reliable tool for numerical simulations of FOWTs in wind-wave environments (Yang et al. (2021); Reig et al. (2024); Rinker et al. (2020)) owing to existing validation efforts using experimental data (Coulling et al. (2013); Robertson et al. (2017)). The OC4 model, in particular, has been validated in OpenFAST against experimental data for free decay tests (Gorostidi et al. (2023); Liu et al. (2019)).

Our simulations aim at solving the system of equations given by:

$$(M + A_\infty)\ddot{q} + Kq = \sum F(t, \omega), \tag{23}$$

where $M$ is the system's mass matrix, $A_\infty$ is the added mass matrix, and $K$ is the hydrostatic stiffness matrix. Terms $q$ and $\ddot{q}$ encompass the system's position and acceleration, respectively (Faltinsen (1993)). The force term can be split into components reflecting the contribution of wave, wind, viscous, mooring, and radiation-damping forces (Jonkman and Matha (2011)).

The simulation process follows the same structure as that presented by Gorostidi et al. (Gorostidi et al. (2023)). We assign the environmental conditions for each simulation by selecting a combination for significant wave height $H_S \in [2, 15]$ $(m)$ and peak period $T_P \in [1, 15]$ $(s)$. The decision to use these features is motivated by their common availability in real practice since Supervisory Control And Data Acquisition (SCADA) systems typically include such measurements.

We have defined evenly-spaced values for both $H_S$ and $T_P$ within their feasible interval, and one combination is randomly selected for each simulation using Monte Carlo sampling. These two variables define a Pierson-Moskowitz spectrum, which estimates the distribution of the energy of ocean waves based on their frequency using the empirical correlation proposed by (Pierson Jr and Moskowitz (1964)). The integration of this spectrum defines the temporal evolution of the wave force component of the total force in Equation 23. We select wind velocity $W_V$ in a similar manner, with speeds ranging from 1 to 30 $m/s$. In this work, we have considered uniform wind speed profiles

We then introduce damage to the mooring lines in MoorDyn, OpenFAST's mooring line dynamics module (Hall (2020)). In this work, as shown in Figure 5, we distort one of the lateral mooring lines of the platform.

Most failures in the mooring system occur during the operational phase, according to a survey focused on FPSOs platforms for the Oil and Gas sector (Fon (2014)). Pitting corrosion, fatigue due to cyclic loading, and abrasion with the seabed may represent some of the most frequent causes leading to critical failure of the mooring systems. In the present study, we analyze the effect of degradation caused by two other common forms of damage: biological fouling (Decurey et al. (2020)) and anchor point slippage (Liu et al. (2021b); Sharma and Nava (2024)). These damage mechanisms affect the mechanical properties of the platform's mooring lines, e.g. mass, stiffness, and buoyancy, and may accelerate wear or cause premature failure (Spraul et al. (2017)). Biofouling, in particular, is a slow process that affects the mass and drag of the mooring system, affecting the performance and stability of the platform. Anchor slipping, moreover, drastically affects the stiffness of the system (Liu et al. (2021a)).

We simulate *biofouling* by modifying the mass per unit length and diameter of a segment located at the center of the mooring line. We consider the maximum biofouling damage to increase these properties by 10%. We induce *anchoring* damage by displacing the line's anchor points $x$ and $y$ coordinates, which we assume to have an effect on the line's stiffness, causing alterations in the response of the platform. We consider the maximum anchoring damage at 20 m in parallel to its baseline orientation of 240 with respect to the $x$ direction. To label the scenarios, we employ two severity coefficients, one for each damage type, where the maximum damage level corresponds to a value equal to one, and healthy mooring line scenarios are assigned null coefficient values. Any intermediate damage coefficient reflects mild degradation, which modifies the line's properties following a linear interpolation. We sample these coefficients by sampling from a folded Gaussian distribution

located around zero, with a standard deviation of 0.35. The rationale behind this is that we seek a training dataset containing mostly low-severity scenarios, much more frequent in common operation (Gorostidi et al. (2023)).

    For any scenario, we first sample the damage coefficients describing the damage condition. We subsequently use Monte Carlo to sample the loading conditions and assign different sea states. With this approach, we build a large dataset with cases that include milder or more energetic sea states for all the considered damage scenarios. Each scenario involves a simulation

recreating 30 minutes of FOWT dynamics, computing platform responses every 0.025 seconds, and subsequently extracting the selected features. In summary, any scenario is defined by (i) three features describing the loading conditions, namely $H_S$, $T_P$, and $W_V$, (ii) 15 response features, five for each of the three considered DOFs, and (iii) the two damaged condition coefficients. We produce 60,000 samples in parallel batches using 120 Intel Xeon (R) E5-2680, 2.70GHz CPUs (Donostia International Physics Centre (2022)), taking approximately 42 hours.

## 4   Neural Network design and training

We employ TensorFlow 2.13 to treat the datasets and train the Bayesian DNN for damaged condition assessment (Abadi et al. (2015)). We split our dataset into training ($\mathcal{D}_{train}$), validation ($\mathcal{D}_{val}$), and testing ($\mathcal{D}_{test}$), each containing 70, 20, and 10% of

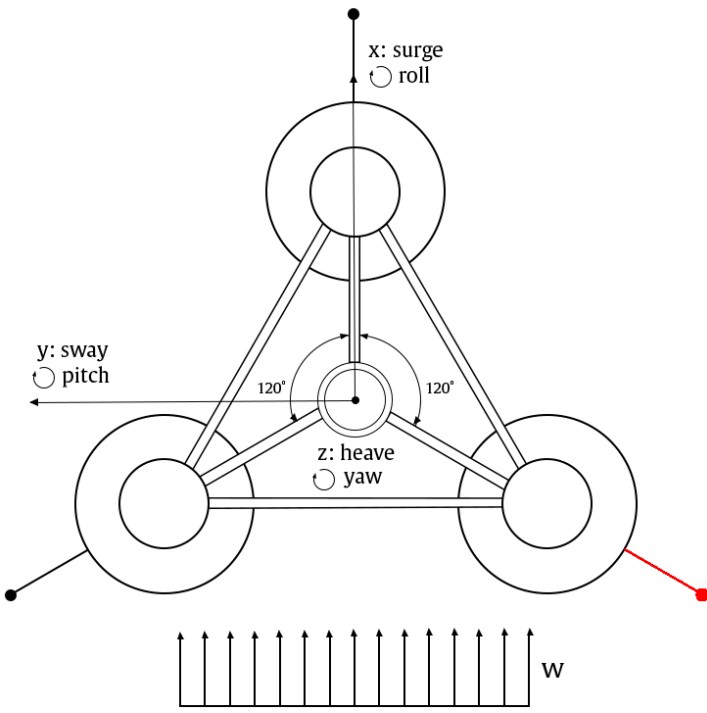

**Figure 5.** Top view of the OC4-DeepCWind semi-submersible platform. The mooring line exposed to damage is highlighted in red.

the total samples, respectively. We then use the MinMax scaler (Scikit Learn (2024)) to constrain the environmental conditions and response features to the interval $[0, 1]$ so as to ensure that high-order features do not outweigh lower-order ones in the training process. The rescaling function is based on the training data and applies to the three datasets.

Next, we specify the DNN architecture and the hyperparameters describing the training routine. There are no general guidelines regulating the pursuit of an optimal configuration. Numerous hyperparameters exist in this case, pertaining to the model definition and its training process, including layer counts, types, and sizes, activation functions, and regularization techniques, among others. This renders the search space virtually infinite. As a common practice, developers usually test the performance of a range of candidate architectures designed on educated guesses and experience, aiming to strike a balance between computational efficiency and prediction accuracy.

In this work, we employ a combination of hyperbolic tangent (Namin et al. (2009)) and Rectified Linear Unit (ReLU) (Agarap (2018)) functions for the hidden layers. We implement the weight initialization method proposed by Aldirany et al. (Aldirany (2024)), who suggested TensorFlow's default Glorot Uniform initialization scheme (Glorot and Bengio (2010)) to be unsuitable for non-differentiable and non-zero mean functions, such as ReLU. Instead, we apply the He Uniform initialization (He et al. (2015)) to the ReLU layers. At the output layer, we use three different activations for the properties of the multivariate Gaussian Mixture: the *sigmoid* function (Han and Moraga (1995)) for the means, as we seek for a smooth function to estimate damaged condition coefficients in the interval $[0, 1]$; the *softplus* function (Zheng et al. (2015)) for the variances, as a smooth equivalent of *ReLU* to enforce positive values; and the *softmax* function (Goodfellow et al. (2016)) for the weights, so that their sum is equal to one and each value ranges into $[0, 1]$. We have observed adequate training performance when using the parameters shown in Table 1. However, other configurations may also provide satisfactory results. For the Multivariate Gaussian Mixture PDF, we assign $k = 5$ Gaussian components. We have no prior knowledge of the optimal number of Gaussians to reflect the uncertainty of the damage condition properties. The larger the number, the more flexible the mixture will be. However, the number of Gaussians directly affects the number of parameters to be estimated; thus, it hampers the training process. After some trial and error analysis, the authors reached adequate results using five Gaussians. We draw $H = 10$ samples to be fed into $\mathcal{F}_{\phi^*}$. A larger number of samples accelerates convergence at the cost of more time-consuming iterations. Finally, we consider a noise parameter $\beta$ equal to 0.076, which roughly corresponds to noise levels of up to approx. $8.5\%$ (Bishop (2006)). The value of $\beta$ is critical to leverage the contribution of both loss terms in Eq.16 during training and thus to properly describe the uncertainty in the solutions.

We follow the two-step training procedure described in Section 2. We first find the optimal forward operator $\mathcal{F}_{\phi^*}$ by minimizing Eq. 3, as described in our previous work (Gorostidi et al. (2023)). We pre-train the decoder to approximate the physical law of the system, since this enables access to the derivative quantities that are needed when training the encoder, according to the loss function in Eq.16. In this manner, the decoder serves to impose the known physical law as a type of inductive bias. A further benefit of this approach is that by specifying fewer unknowns (only those corresponding to the encoder), we reduce the difficulty of the inference task; in particular, we decrease the number of local minima.

| Encoder | |
| --- | --- |
| Layers | 100, 250, 300, 300, 200, 150, 100 |
| Activations | ReLU, ReLU, Tanh, ReLU, Tanh, ReLU, Tanh |
| Weight init. | GU, GU, HU, GU, HU, GU, HU |
| Initial LR | $10^{-5}$ |
| Batch size | 1024 |
| Epochs | 200 |
| **Sampling layer** | |
| $\mu$ activation | Sigmoid |
| $\sigma$ activation | Softplus |
| $w$ activation | Softmax |
| Num. Gaussians | 5 |
| Num. samples | 10 |
| **Decoder** | |
| Layers | 10, 30, 50, 70, 80 |
| Activations | Tanh, ReLU, ReLU, ReLU, ReLU |
| Weight init. | GU, HU, HU, HU, HU |
| Initial LR | $5 \cdot 10^{-3}$ |
| Batch size | 512 |
| Epochs | 500 |

**Table 1.** Specifications of our Gaussian Mixture autoencoder. Tanh: Hyperbolic Tangent; ReLU: Rectified Linear Unit; GU: Glorot Uniform, HU: He Uniform.

We present the evolution of the decoder's loss in Figure 6. The parameters of $\mathcal{F}_{\phi^*}$ are frozen for the next training step. We obtain the optimal inverse operator $\mathcal{I}_{\theta^*}$ by minimizing the ELBO loss described in Eq. 16. Figure 7 depicts the evolution of the loss function during training, including the total loss value, and the two participating terms.

## 5   Results with three DOFs

This section analyzes the damage identification performance using the test dataset $\mathcal{D}_{test}$ unseen during training and validation tasks. Here, we select eight damage scenarios from $\mathcal{D}_{test}$ to visualize the results. The eight data points represent different damaged conditions and are summarized in table 2.

The deterministic approach employed in (Gorostidi et al. (2023)) minimized the data misfit ($\mathcal{L}_{\mathcal{M}}$), producing a single value estimate for each input measurement. However, despite being a feasible solution, it might be far from the true one when multiple solutions coexist. For any response measurement $\mathbf{m}$, we can identify the feasible solutions as those producing a

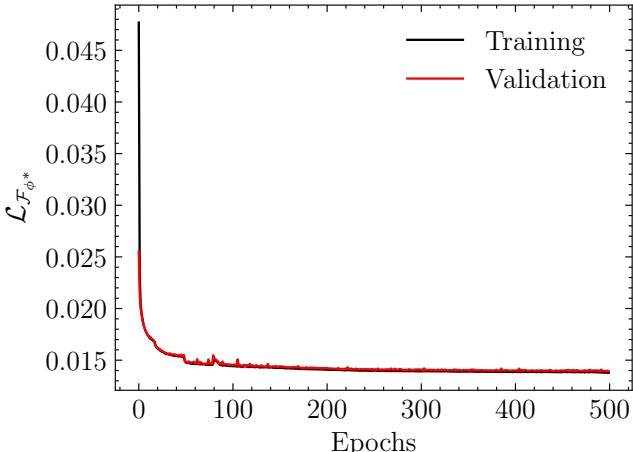

**Figure 6.** Evolution of training and validation losses for the forward operator $\mathcal{F}_{\phi^*}$

**Table 2.** Properties of the selected examples

| Scenario | $z_1$ | $z_2$ |
|----------|-------|-------|
| 1 | 0.20 | 0.65 |
| 2 | 0.82 | 0.44 |
| 3 | 0.34 | 0.74 |
| 4 | 0.00 | 0.00 |
| 5 | 0.15 | 0.60 |
| 6 | 0.05 | 0.25 |
| 7 | 0.34 | 0.10 |
| 8 | 0.00 | 0.40 |

reduced value of $\mathcal{L}_{\mathcal{M}}$. Accordingly, these solutions must produce a high probability value in the estimated posterior PDF, $q_{\boldsymbol{\theta}}(\mathbf{z}|\mathbf{m}, \mathbf{w})$. Figures 8 and 9 depict the solution space comparing two contour maps within the solution space: (i) the density value over $q_{\boldsymbol{\theta}}(\mathbf{z}|\mathbf{m}, \mathbf{w})$, and (ii) the $\mathcal{L}_{\mathcal{M}}$ value for the eight test example scenarios. The contour plot of data misfit reveals the existence of multiple damage scenarios producing similar responses (measurements), i.e., producing a small value of $\mathcal{L}_{\mathcal{M}}$. Both contours must ideally be identical for the posterior to show all the feasible solutions. However, the constraints and assumptions imposed for the posterior to be tractable (parametrization of the distribution to a Gaussian mixture, diagonal covariance matrix assumption, etc.) restrict the shape of the PDF. For comparison purposes, the figures include both the ground truth and the deterministic solutions produced in work (Gorostidi et al. (2023)).

In the figures, we have constrained the contour maps of $\mathcal{L}_{\mathcal{M}}$ to enable proper visualization of the targeted regions. Despite the shape limitations of the estimated posterior distributions, the resulting contours (left-hand figures) exhibit a clear correspon-

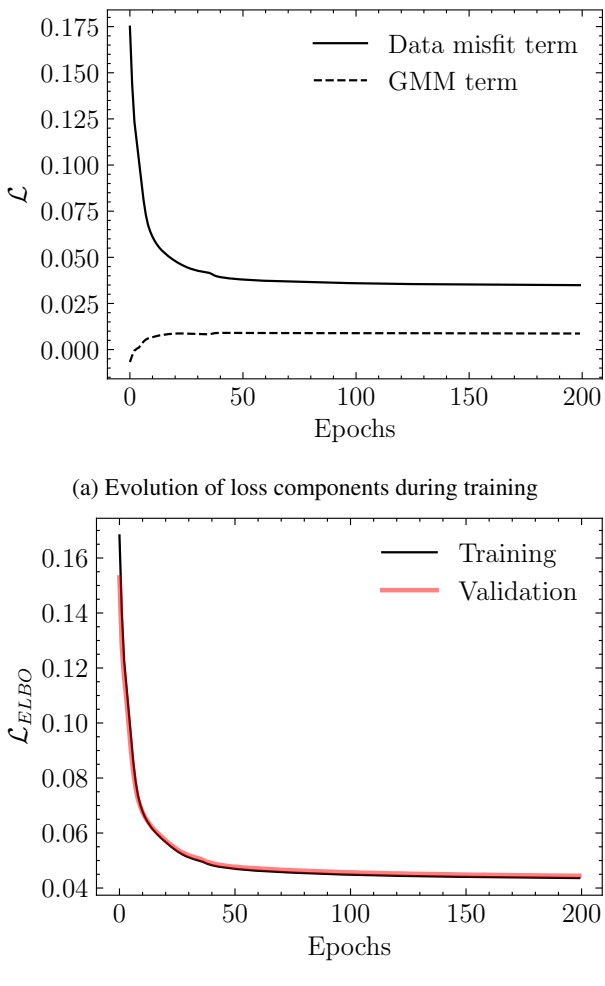

(a) Evolution of loss components during training

(b) ELBO loss during training

**Figure 7.** Evolution of training and validation losses for the 3-DOF case: (a) data misfit and mixture density components of the loss, and (b) training and validation ELBO losses

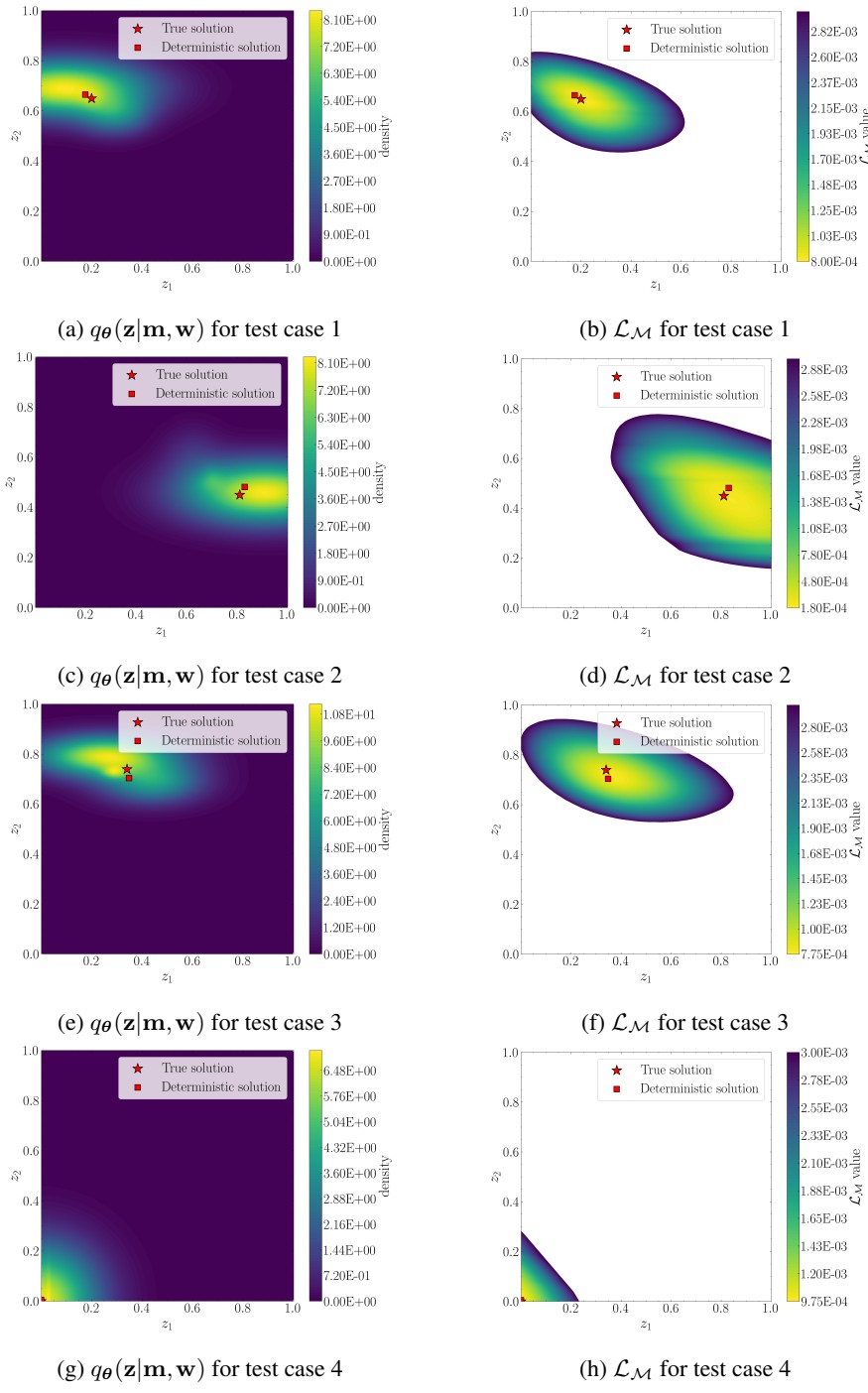

**Figure 8.** First four test examples: The left-hand figures represent the contour plot of the estimated posteriors $q_{\boldsymbol{\theta}}(\mathbf{z}|\mathbf{m}, \mathbf{w})$. The right-hand figures represent the data misfit ($\mathcal{L}_{\mathcal{M}}$) value.

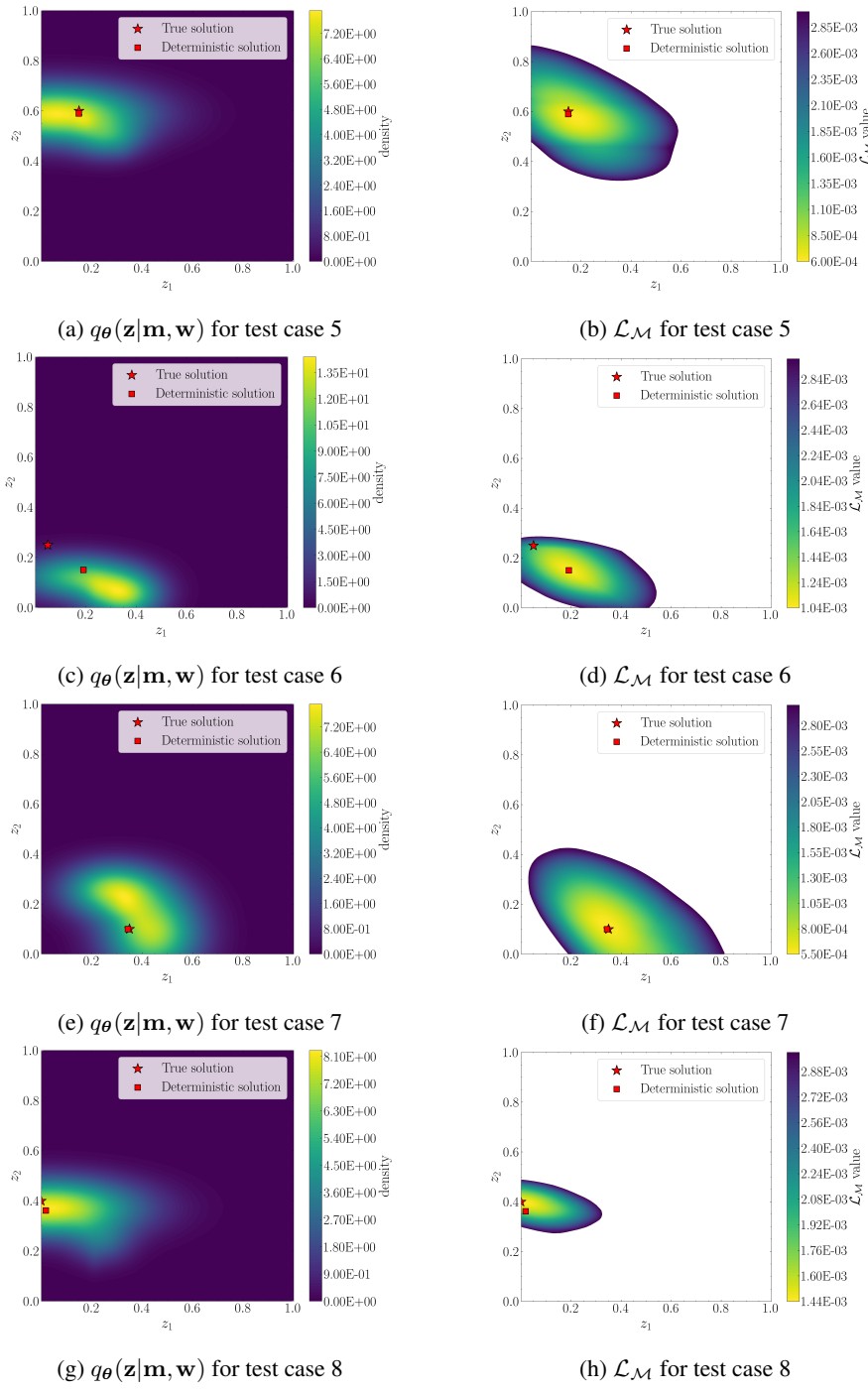

(a) $q_{\theta}(\mathbf{z}|\mathbf{m}, \mathbf{w})$ for test case 5

(b) $\mathcal{L}_{\mathcal{M}}$ for test case 5

(c) $q_{\theta}(\mathbf{z}|\mathbf{m}, \mathbf{w})$ for test case 6

(d) $\mathcal{L}_{\mathcal{M}}$ for test case 6

(e) $q_{\theta}(\mathbf{z}|\mathbf{m}, \mathbf{w})$ for test case 7

(f) $\mathcal{L}_{\mathcal{M}}$ for test case 7

(g) $q_{\theta}(\mathbf{z}|\mathbf{m}, \mathbf{w})$ for test case 8

(h) $\mathcal{L}_{\mathcal{M}}$ for test case 8

**Figure 9.** Last four test examples: The left-hand figures represent the contour plot of the estimated probability density functions $q_{\theta}(\mathbf{z}|\mathbf{m}, \mathbf{w})$. The right-hand figures represent the data misfit ($\mathcal{L}_{\mathcal{M}}$) value.

dence with the expected solution space described by $\mathcal{L}_{\mathcal{M}}$. Thus, the contour plot of the estimated posterior PDF $q_{\boldsymbol{\theta}}(\mathbf{z}|\mathbf{m}, \mathbf{w})$ describes the uncertainty in the solution space.

We observe that, in most cases, the shape accommodates close to a bi-variate Gaussian distribution, indicating the existence of a dominating mode. These results suggest that the selected sensing system and the extracted features suffice to uniquely identify the damaged condition causing the observed measurements. Analyzing the deterministic solution, we observe that it lies close to the true damaged condition, indicating acceptable predictions. However, as we will explore in subsection 6, employing an incomplete sensing system (fewer DOFs available) might contribute to the ill-posedness of the inverse and thus result in a much more uncertain damaged condition.

In this work, we have made two main assumptions in the training dataset to constrain the scope of the analysis. On the one hand, the results presented in this section assumed the availability of some loading condition features describing the wind and wave excitation. Although SCADA systems are frequently found in FOWTs, such measurements may be inaccessible or be subject to failures. Appendix A describes the results obtained when training the inverse operator without any information regarding the loading conditions.

On the other hand, we have particularized the methodology to the damaged condition of one single mooring line. As a proof of concept, we assume that only one mooring line may suffer damage at a time. However, in real practice, simultaneous damage may happen in two or more components of the target system (e.g., two mooring lines). Appendix B presents the results obtained for a new case study where we consider anchoring damage co-occurring at two mooring lines.

Given the computational limitations of simulating damage scenarios, a relevant aspect that needs to be considered relates to the capability of the DNN to provide a reliable outcome for a certain scenario that was unseen during training (although of the same nature/type as those used for training). Appendix C analyzes such a situation via two different studies to demonstrate the ability of the distributional model to approximate the true solution and inform about the uncertainty origin.

## 5.1 Exploring robustness to noise

One critical issue to tackle when dealing with data from real operative systems is the effect of noise. We represent the measurement error as an additive noise with Gaussian distribution. We assume the covariance matrix to be a diagonal matrix $\Gamma = (\beta \mathcal{F}([\mathbf{z}, \mathbf{w}]))^2$, where the scalar $\beta$ affects the variances of the noise components. A low value of $\beta$ indicates almost null noise in the data, and the total loss value is mainly owed to the data misfit term. Contrarily, a larger value of $\beta$ is associated with higher noise levels and an increase in the contribution of the distributional term. We can relate $\beta$ with the signal-to-noise ratio (SNR), which is given by (see (Johnson (2006))):

$$SNR = 10 log_{10} \frac{\mathbb{E}(S^2)}{\mathbb{E}(n^2)}, \tag{24}$$

where $S$ represents the measured signal, and $n$ the noise. For white Gaussian noise with a null mean, we substitute the expectation of the noise by its variance. As an indicative value of the noise level based on the training dataset, assigning $\beta = 0.05$ corresponds to a $SNR \approx 26 dB$.

We explore increasing noise levels by modifying the value of $\beta$. Figure 10 depicts this effect for the first test example in Table 2. Analogous results are observed for the other examples. Results reveal the sensitivity of the estimated PDF to the noise

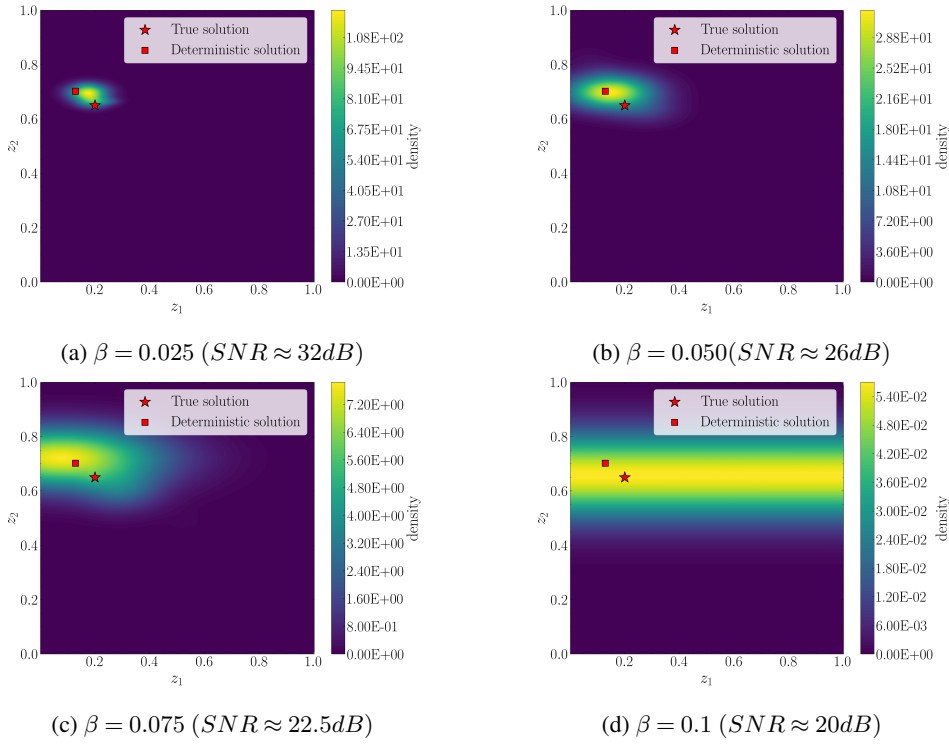

(a) $\beta = 0.025$ ($SNR \approx 32dB$)

(b) $\beta = 0.050$ ($SNR \approx 26dB$)

(c) $\beta = 0.075$ ($SNR \approx 22.5dB$)

(d) $\beta = 0.1$ ($SNR \approx 20dB$)

**Figure 10.** Analysis of noise effect in the $q_{\boldsymbol{\theta}}(\mathbf{z}|\mathbf{m}, \mathbf{w})$ results for the first test example. Increasing noise levels according to $\beta = [0.025, 0.050, 0.075, 0.10]$

level. We observe that for $\beta = 0.1$ ($SNR \approx 20dB$), the entire solution space is feasible for the biofouling damage represented by $z_1$. This demonstrates a limited sensitivity of the measured features to this damage compared to anchoring ($z_2$), which can still be estimated. These results support the robustness of the method in the presence of high noise levels.

## 6   Results with one DOF

To highlight the potency of the proposed method, we explore the multimodality of the solution space when the instrumentation
system is sparse (limited number of sensors). This situation is very common when low-cost, long-term sensing devices are installed. Measuring accelerations is also an extended and cost-effective practice. Damage in the mooring elements affects the dynamics of the entire FOWT system (including the six degrees of freedom). However, the dynamics of the mooring system correspond to low frequencies and strongly exciting sway DOF, which is coupled with the rotation DOF roll. Compared to accelerations, displacements (and coupled rotation DOFs) tend to be more sensitive to damage in the mooring systems, such
as anchor displacements, in terms of signal power (Sharma and Nava (2024)). To explore the multimodality in the damaged

condition estimates, we have selected roll DOF as the only available signal since it is particularly sensitive to damage given the mechanical symmetry of the system and the unidirectionally of the external excitation, and aids in the visualization of the results.

In this case, we employ only five response features - those associated with roll - as the input data for the DNN. We adapt the architecture of $\mathcal{I}_{\boldsymbol{\theta}}$ described in Section 4 to accommodate the new input dimensions and repeat the training step to minimize $\mathcal{L}_{ELBO}$. Figures 11 and 12 show the results for the eight damage scenarios.

The data misfit ($\mathcal{L}_{\mathcal{D}}$) contour maps on the right-hand side of Figure 12 reveal a much more spread range of plausible damaged conditions across the solution space for any given input measurement. The contour plots of the estimated probabilities (left-hand figures) match with the $\mathcal{L}_{\mathcal{D}}$ map. Given that the uncertainty has increased in the solution space (ill-posedness), the results reveal a more clear multimodality in the solution space. Compared to Figures 8 and 9, where the deterministic solution was pretty close to the ground truth, now we observe a considerable separation. This discrepancy already informs on the existence of multiple solutions. For example, for the test case 1 (see Figures 11a and 11b), we clearly observe how the Gaussians try to adopt the tube-shaped contour map of minimal $\mathcal{L}_{\mathcal{D}}$ values that seem to connect the ground truth and the deterministic solution.

Figures 9 through 12 qualitatively illustrate the multimodal nature of the solution to our inverse problem. Next, we further quantify the performance of our network using common metrics. Table 3 reports the root mean square error (RMSE) of the estimations using both deterministic and probabilistic models. The table's left half presents the RMSE values for the 3-DOF case, while its right half displays the RMSEs for the case of incomplete instrumentation with roll DOF only. RMSEs were calculated using ten samples from the Gaussian mixture autoencoder. Since the deterministic model provides a point prediction, the RMSEs of the deterministic predictions are equivalent to the absolute error between the ground truth and the model's estimation.

The 3-DOF errors in Table 3 concur with the contour plots presented in Figures 8 and 9, which proved the selected instrumentation sufficient for the deterministic model to accurately estimate mooring degradation. Table 3 shows that the deterministic model's predictions using 1-DOF are poorer, with our Bayesian approach exhibiting improved behavior, especially in Scenarios 1, 3, and 7. These cases, as shown in Figures 11 and 12, reflect a highly multimodal solution and prove that the deterministic model does not accurately estimate the true solution but rather captures one of the plausible solutions.

These results support the capacity of the proposed method to provide an uncertainty-aware assessment in the form of the probability value for each pair of damaged conditions. In such cases, postprocessing can be applied to extract the relevant information from the multivariate contour map, such as defining extrema or establishing a threshold to select the most likely scenarios. Statistical data compression techniques (e.g., Principal Component Analysis (PCA)) could be applied to produce a lower-dimensional representation of the space that enables visualization.

## 7 Conclusions

This work proposes a Bayesian Deep Learning strategy to achieve an uncertainty-aware assessment of the condition of the mooring system in a Floating Offshore Wind Turbine (FOWT). To remedy this, we employ a mixture of multivariate Gaussians

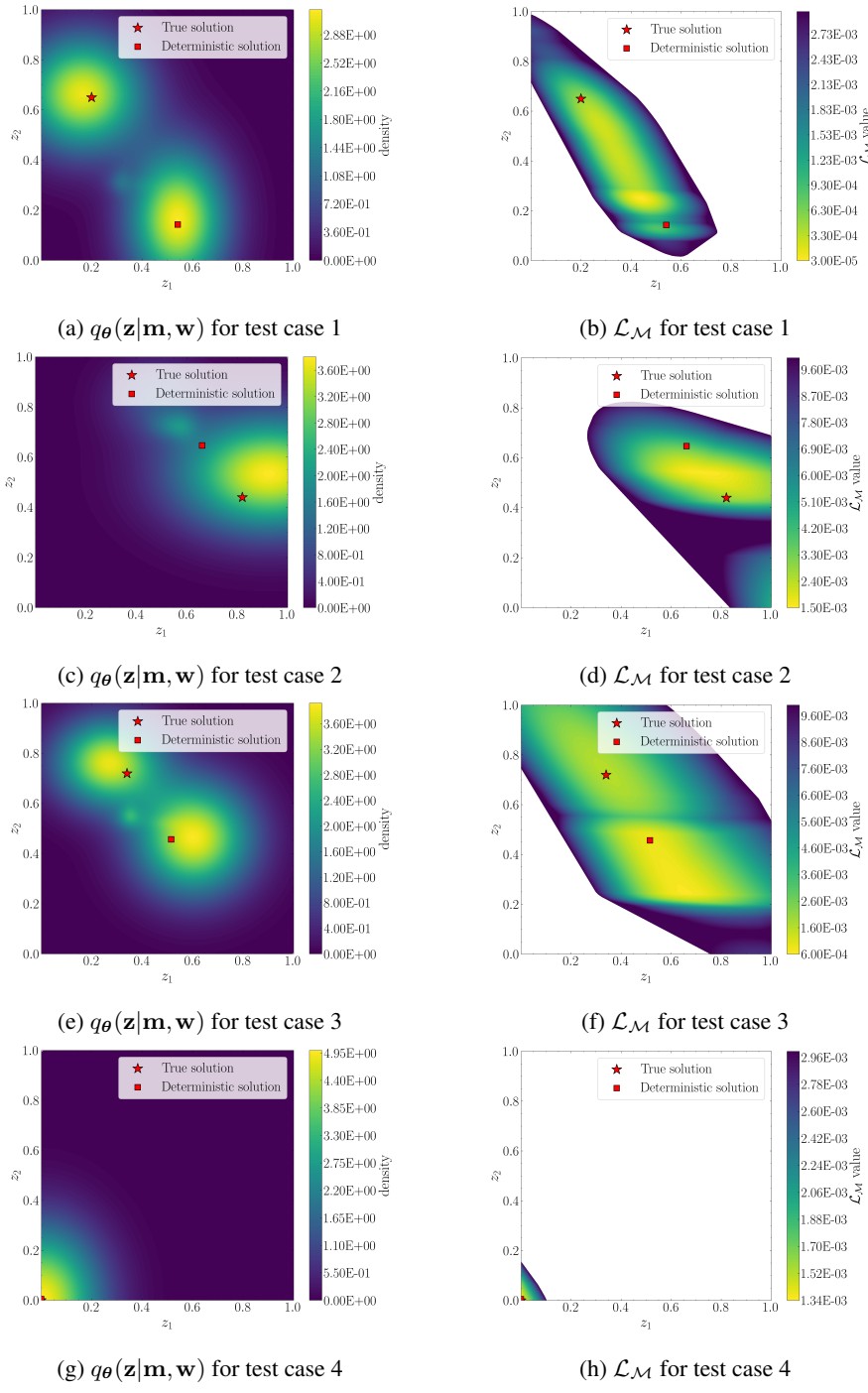

(a) $q_\theta(\mathbf{z}|\mathbf{m},\mathbf{w})$ for test case 1

(b) $\mathcal{L}_\mathcal{M}$ for test case 1

(c) $q_\theta(\mathbf{z}|\mathbf{m},\mathbf{w})$ for test case 2

(d) $\mathcal{L}_\mathcal{M}$ for test case 2

(e) $q_\theta(\mathbf{z}|\mathbf{m},\mathbf{w})$ for test case 3

(f) $\mathcal{L}_\mathcal{M}$ for test case 3

(g) $q_\theta(\mathbf{z}|\mathbf{m},\mathbf{w})$ for test case 4

(h) $\mathcal{L}_\mathcal{M}$ for test case 4

**Figure 11.** First four test examples for one DOF: The left-hand figures represent the contour plot of the estimated probability density functions $q_\theta(\mathbf{z}|\mathbf{m},\mathbf{w})$. The right-hand figures represent the data misfit ($\mathcal{L}_\mathcal{M}$) value.

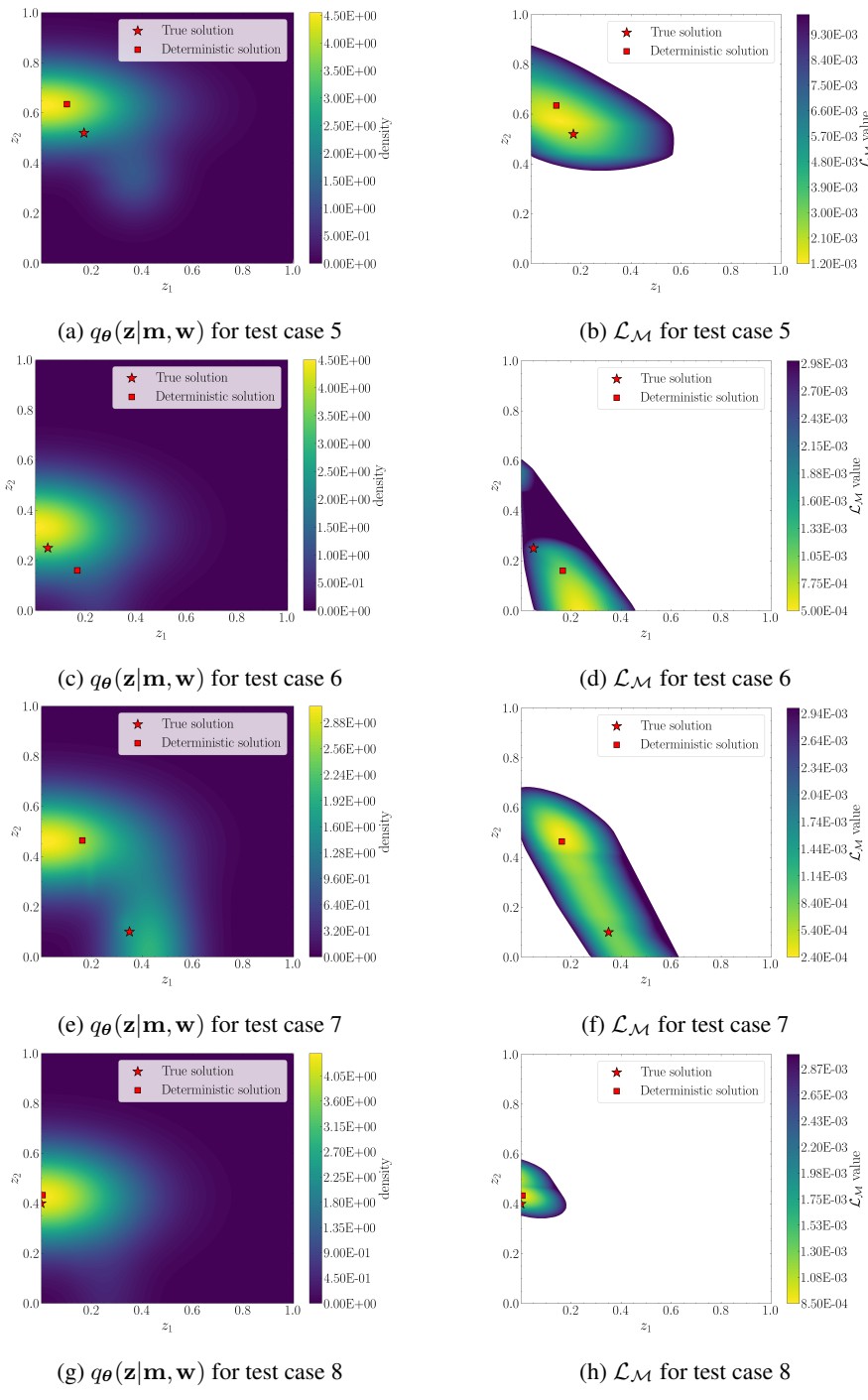

(a) $q_{\boldsymbol{\theta}}(\mathbf{z}|\mathbf{m},\mathbf{w})$ for test case 5

(b) $\mathcal{L}_{\mathcal{M}}$ for test case 5

(c) $q_{\boldsymbol{\theta}}(\mathbf{z}|\mathbf{m},\mathbf{w})$ for test case 6

(d) $\mathcal{L}_{\mathcal{M}}$ for test case 6

(e) $q_{\boldsymbol{\theta}}(\mathbf{z}|\mathbf{m},\mathbf{w})$ for test case 7

(f) $\mathcal{L}_{\mathcal{M}}$ for test case 7

(g) $q_{\boldsymbol{\theta}}(\mathbf{z}|\mathbf{m},\mathbf{w})$ for test case 8

(h) $\mathcal{L}_{\mathcal{M}}$ for test case 8

**Figure 12.** Last four test examples for one DOF: The left-hand figures represent the contour plot of the estimated probability density functions $q_{\boldsymbol{\theta}}(\mathbf{z}|\mathbf{m},\mathbf{w})$. The right-hand figures represent the data misfit ($\mathcal{L}_{\mathcal{M}}$) value.

| | 3-DOFs | | | | 1-DOF | | | |
|---|---|---|---|---|---|---|---|---|
| | Deterministic | | Gaussian Mixture | | Deterministic | | Gaussian Mixture | |
| Scenario | $\text{RMSE}_{z_1}.$ | $\text{RMSE}_{z_2}$ | $\text{RMSE}_{z_1}$ | $\text{RMSE}_{z_2}$ | $\text{RMSE}_{z_1}.$ | $\text{RMSE}_{z_2}$ | $\text{RMSE}_{z_1}$ | $\text{RMSE}_{z_2}$ |
| 1 | 0.011 | 0.013 | 0.114 | 0.041 | 0.355 | 0.494 | 0.330 | 0.357 |
| 2 | 0.010 | 0.023 | 0.133 | 0.062 | 0.133 | 0.062 | 0.131 | 0.107 |
| 3 | 0.006 | 0.004 | 0.181 | 0.148 | 0.180 | 0.183 | 0.149 | 0.185 |
| 4 | 0.053 | 0.002 | 0.103 | 0.032 | 0.052 | 0.051 | 0.151 | 0.153 |
| 5 | 0.002 | 0.005 | 0.200 | 0.093 | 0.021 | 0.090 | 0.137 | 0.118 |
| 6 | 0.141 | 0.071 | 0.161 | 0.136 | 0.126 | 0.077 | 0.112 | 0.180 |
| 7 | 0.095 | 0.004 | 0.091 | 0.135 | 0.164 | 0.374 | 0.123 | 0.210 |
| 8 | 0.012 | 0.020 | 0.210 | 0.066 | 0.014 | 0.034 | 0.046 | 0.018 |

**Table 3.** RMSE values for 3-DOF and 1-DOF cases using both deterministic and Gaussian mixture autoencoder models.

to track how the uncertainty is propagated from the available measurements to the estimated health condition diagnostics,
providing more robust and reliable estimates. We test the performance of the method using measurements from three degrees of freedom and explore the robustness against increasing noise levels, with successful results. We also analyze the benefits of the method when dealing with sparse sensor scenarios, such as measuring at only one degree of freedom, revealing the ability of the method to reveal the multimodal nature of the solution in ill-posed scenarios.

In line with the limitations and challenges stated in Section 1, we determine here future research lines and challenges to be addressed. Scaling up the methodology to more complex damage spaces (e.g., simultaneous damage in all mooring lines, different directions of the anchor displacements, corrosion effect) is one of the key challenges to be addressed. Given the intractable dimensionality increase in the number of parameters of the Gaussian mixture, we will explore alternate distributional models, such as random fields (Birmpa and Katsoulakis (2021)), Gaussian Processes (GPs) (Li et al. (2019)), or Copulas and vine Copulas (Letizia and Tonello (2022)).

We further consider a future research line to focus on more exhaustive feature extraction that includes more refined system identification features, including the transmissibility and transmittance, or autocorrelation features, as they may enhance the damage identification task. Also, to achieve a higher level assessment, we will investigate clustering techniques to separate the different types of uncertainty according to the nature and properties of the existing sources (e.g., multimodality of the estimate, measurement noise, model uncertainties, extrapolation over unknown measurements). Kamariotis et al. (2024).

Finally, it is a core priority of the authors to validate the proposed methodology for a case study where both experimental and synthetic data can be combined. Experimental data will present important limitations, including a strong concentration around a reduced region of the domain (mostly the undamaged condition and other unlabeled damage scenarios). Hence, it is mandatory to find an adequate strategy to complement these data with synthetic simulations, accounting for the computational limitations in terms of time and resources. In this sense, covering the entire range of possible damage scenarios is computa-

535 tionally prohibitive unless it is carefully tackled. Active Learning methods enable an efficient design of the synthetic dataset required for the training stage, as they guide the simulation process to enrich the most poorly characterized regions.

Finally, given the potential of the proposed methodology to be applied to different inverse problems, the authors are currently exploring its application to the field of damage identification for bridge structures. The main challenges arising in this context relate to the high dimensionality of the damage condition space, which will require more efficient distributional models such
as copulas.

*Code availability.*   The code is available in the GitHub repository of the MATHMODE group (https://github.com/Mathmode/GMM_Autoencoder.git)

*Author contributions.*   Ana Fernandez-Navamuel participated in the Methodology development and investigation phases and was in charge of the writing phase (original draft preparation, reviewing, and editing). She also contributed to funding acquisition. Nicolas Gorostidi
participated in the data curtation step, the methodology, and the original draft preparation. David Pardo participated in the conceptualization process, methodology, and investigation phases as supervisor and in the review and editing of the manuscript. He also acquired financial support and resources to properly develop the present work. Vincenzo Nava participated in the conceptualization process, supervision, and writing (reviewing and editing). He also provided funding resources. Eleni Chatzi participated in the investigation and validation stages. She also contributed to writing (review and editing) and obtaining funding support to develop this manuscript.

*Competing interests.*   The authors declare that there are no competing interests.

*Acknowledgements.*   The authors would like to acknowledge the ELKARTEK program (TCRINI project (KK-2023-0029), BEREZ-IA (KK-2023/00012), RUL-ET (KK-2024/00086), and SEGURH2 (KK-2024/00068)), the BERC 2022–2025 program, and the Consolidated Research Group MATHMODE (IT1456-22), given by the Department of Education and the BCAM-IKUR-UPV/EHU, funded by the Basque Government IKUR Strategy and by the European Union NextGenerationEU/PRTR, given by the Department of Education, and the grant
PID2023-146678OB-I00 funded by MICIU/AEI/10.13039/ 501100011033 and by the European Union NextGenerationEU/PRTR; the Spanish Ministry of Science and Innovation projects with references TED2021-132783B-I00 funded by MCIN/ AEI /10.13039/501100011033 and the European Union NextGenerationEU/ PRTR; the Spanish Ministry of Economic and Digital Transformation with Misiones Project IA4TES (MIA.2021.M04.008 / NextGenerationEU PRTR); the "BCAM Severo Ochoa" accreditation of excellence CEX2021-001142-S/MICIN/AEI/10.13039/ 501100011033; the European Union's Horizon Europe research and innovation programme with FUTURAL project
(Grant ID: 101083958), LIAISON project (Grant ID 101103698), and the Marie Sklodowska-Curie under (Grant ID 101119556). The authors also wish to acknowledge the MSCA Staff Exchanges 2021 project ReCharged (Grant ID: 101086413). This project has received funding from the Horizon Europe Programme under the Marie Skłodowska-Curie Staff Exchanges Action (GA no. 101086413), co-funded by the UK Research & Innovation and the Swiss State Secretariat for Education, Research & Innovation.

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

**Appendix A:  Exploring results when loading conditions are unavailable**

This appendix considers the case when the loading condition features describing the wind and wave excitation are unavailable. We train our Inverse model $\mathcal{I}_{\boldsymbol{\theta}}$ ignoring these features for the case where only Roll DOF is available. The design and training specifications are similar to those summarized in Table 1 Figure A1 presents the results obtained for the first four test case scenarios. Analyzing the results, we observe that, although the distributional model partially captures the damage condition in some cases (see e.g., Figures A1a A1c A1g), in other scenarios it is far from producing reliable outcomes. The lack of information regarding the loading conditions misleads the condition estimates, mainly for variable $z_1$ (anchoring). For example, in Figures A1b and A1e, we observe that the distributional model spreads along the anchoring axis ($z_1$) with no identification of the existing multimodality.

**Appendix B:  Exploring results for simultaneous damage in two mooring lines**

In this appendix, we analyze the detection capability of simultaneous damage occurring in two different mooring lines. To remain in the two-dimensional space, we consider only anchoring damage. In particular, this section explores anchoring damage to a lateral mooring line, as well as the downstream line parallel to surge motion. The parameterization of anchoring degradation for the platform's lateral line remains unchanged with respect to the main body of this study. Anchoring damage displaces the anchor of the floater's downstream line up to 20 m further in the direction of the wind-wave excitation. Figure B1 presents the probability and data misfit distributions for four test cases, considering as input data the three rotation DOFs (roll, pitch, and

yaw). The figure shows the robust performance of the model to concurrent damage in various mooring lines. However, if we constrain the input data to roll DOF only, we find that the measured features are unable to identify the damage in the second mooring line. Figure B2 illustrates this for one test scenario. The results highlight the importance of selecting damage-sensitive features to capture the different modes of failure that may occur in the system.

## Appendix C: Exploring the generalization capability for unseen damage scenarios

This appendix explores the potential of our proposed method in evaluating damage scenarios different from the training cases. Given the extrapolation limitations of Neural Networks, we study scenarios of the same nature (the same type of damage occurs, namely, anchoring and biofouling), although with a remarkable reduction in severity. We perform two different analyses for the inverse: (i) training only with scenarios where damage $z_1$ (anchoring) is always $\geq 20\%$ and testing with scenarios with $z_1 < 20\%$, and (ii) training only with scenarios where damage $z_2$ (biofouling) is always $\geq 20\%$ and testing with scenarios with $z_2 < 20\%$. In both cases, we consider using the three rotation DOFs as the input measured signals (roll, pitch, and yaw).

Figure C1 shows four randomly selected test cases for the first analysis (anchoring kept $\geq 20\%$ during training). We observe that the distributional model successfully approximates the true solution, with the uncertainty spreading mostly in the direction of the constrained feature (i.e., anchoring) within acceptable thresholds. Hence, the developed method can inform on the damage condition and indicate the uncertainty direction even for cases where the evaluated scenario lies outside the training domain.

Analogously, Figure C2 shows four test scenarios for the second analysis (biofouling kept $\geq 20\%$ during training). The results reveal that the highest density values concentrate close to the true solution, although with some error (see, e.g., Figure ??, where the highest PDF values are shifted towards smaller anchoring damage, but the level of biofouling damage is well captured).

In summary, the results demonstrate that despite these not being included during the training phase, the DNN successfully identifies damage scenarios with lighter severity. The analyses demonstrate the generalizability of the method for damage scenarios that have the same nature but are significantly different in severity from those considered during training, as well as the potential of the distributional model to indicate the direction of the uncertainty.

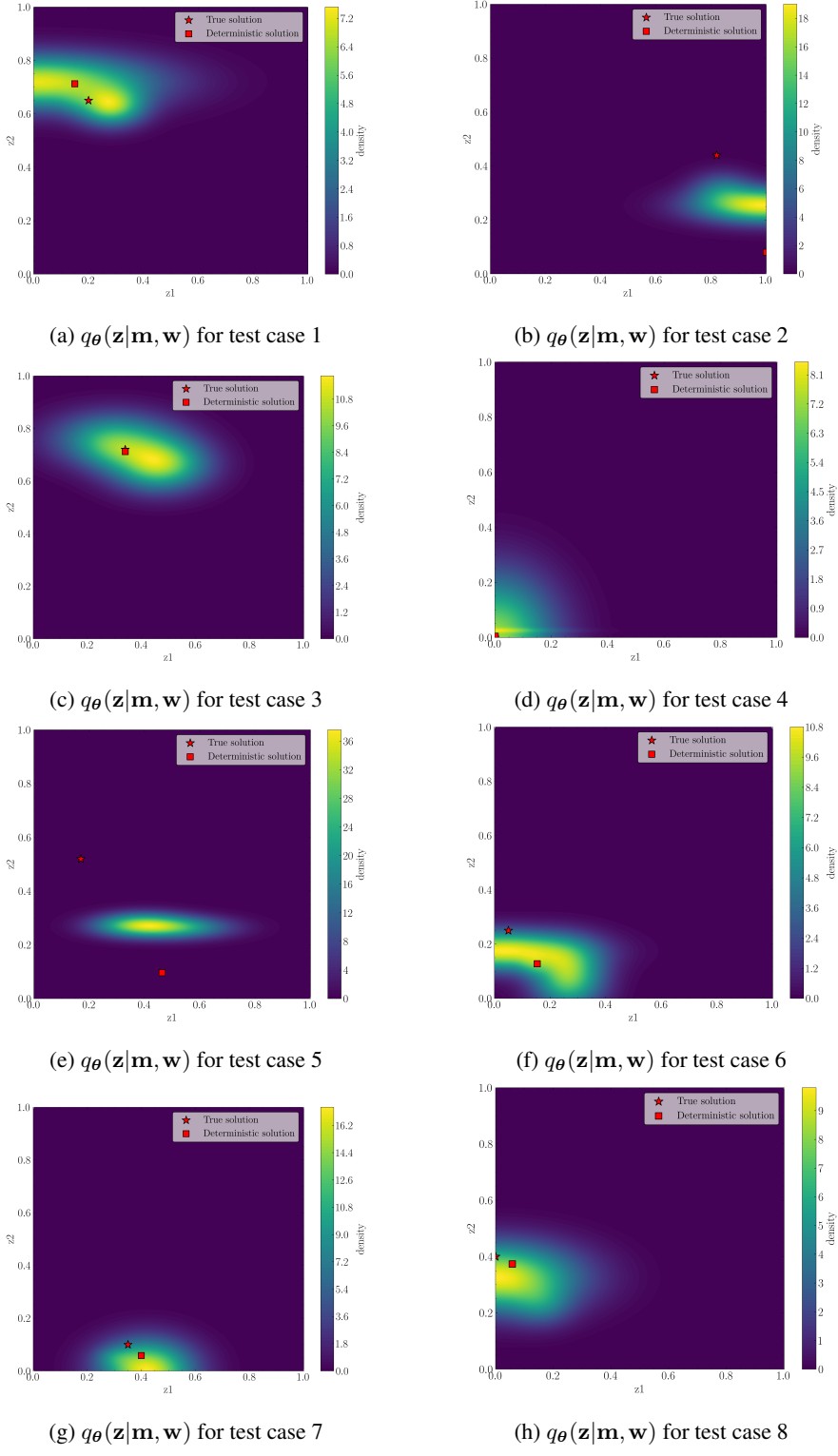

(a) $q_{\boldsymbol{\theta}}(\mathbf{z}|\mathbf{m}, \mathbf{w})$ for test case 1

(b) $q_{\boldsymbol{\theta}}(\mathbf{z}|\mathbf{m}, \mathbf{w})$ for test case 2

(c) $q_{\boldsymbol{\theta}}(\mathbf{z}|\mathbf{m}, \mathbf{w})$ for test case 3

(d) $q_{\boldsymbol{\theta}}(\mathbf{z}|\mathbf{m}, \mathbf{w})$ for test case 4

(e) $q_{\boldsymbol{\theta}}(\mathbf{z}|\mathbf{m}, \mathbf{w})$ for test case 5

(f) $q_{\boldsymbol{\theta}}(\mathbf{z}|\mathbf{m}, \mathbf{w})$ for test case 6

(g) $q_{\boldsymbol{\theta}}(\mathbf{z}|\mathbf{m}, \mathbf{w})$ for test case 7

(h) $q_{\boldsymbol{\theta}}(\mathbf{z}|\mathbf{m}, \mathbf{w})$ for test case 8

**Figure A1.** First four test examples for one DOF when omitting wind and wave condition measurements in the inverse training stage.

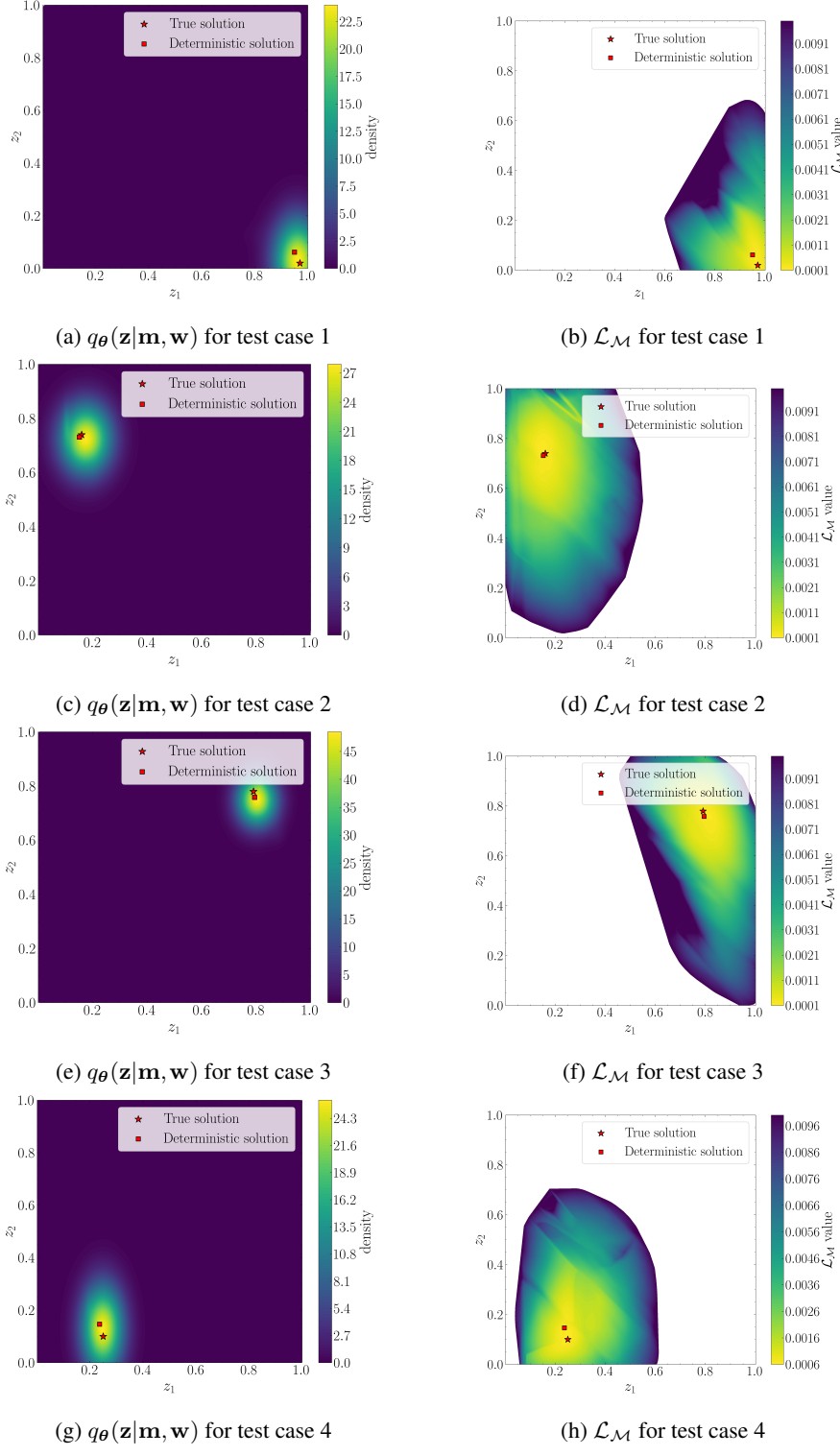

(a) $q_{\boldsymbol{\theta}}(\mathbf{z}|\mathbf{m},\mathbf{w})$ for test case 1

(b) $\mathcal{L}_{\mathcal{M}}$ for test case 1

(c) $q_{\boldsymbol{\theta}}(\mathbf{z}|\mathbf{m},\mathbf{w})$ for test case 2

(d) $\mathcal{L}_{\mathcal{M}}$ for test case 2

(e) $q_{\boldsymbol{\theta}}(\mathbf{z}|\mathbf{m},\mathbf{w})$ for test case 3

(f) $\mathcal{L}_{\mathcal{M}}$ for test case 3

(g) $q_{\boldsymbol{\theta}}(\mathbf{z}|\mathbf{m},\mathbf{w})$ for test case 4

(h) $\mathcal{L}_{\mathcal{M}}$ for test case 4

**Figure B1.** Four test cases for simultaneous anchoring damage on two mooring lines considering roll, pitch, and yaw motion.

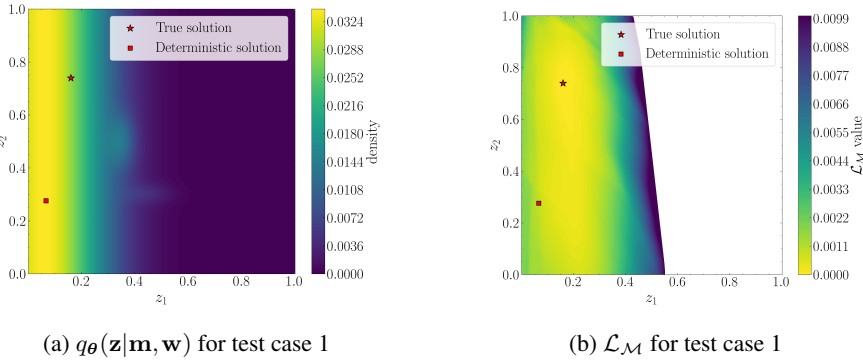

(a) $q_{\boldsymbol{\theta}}(\mathbf{z}|\mathbf{m}, \mathbf{w})$ for test case 1

(b) $\mathcal{L}_{\mathcal{M}}$ for test case 1

**Figure B2.** Test case for simultaneous anchoring damage on two mooring lines considering only roll motion.

![Posterior PDF contour plots for four test scenarios]

**Figure C1.** Posterior PDF ($q_{\boldsymbol{\theta}}(\mathbf{z}|\mathbf{m}, \mathbf{w})$) contour plots for four test scenarios when training with anchoring ($z_1$) scenarios $\geq 20\%$

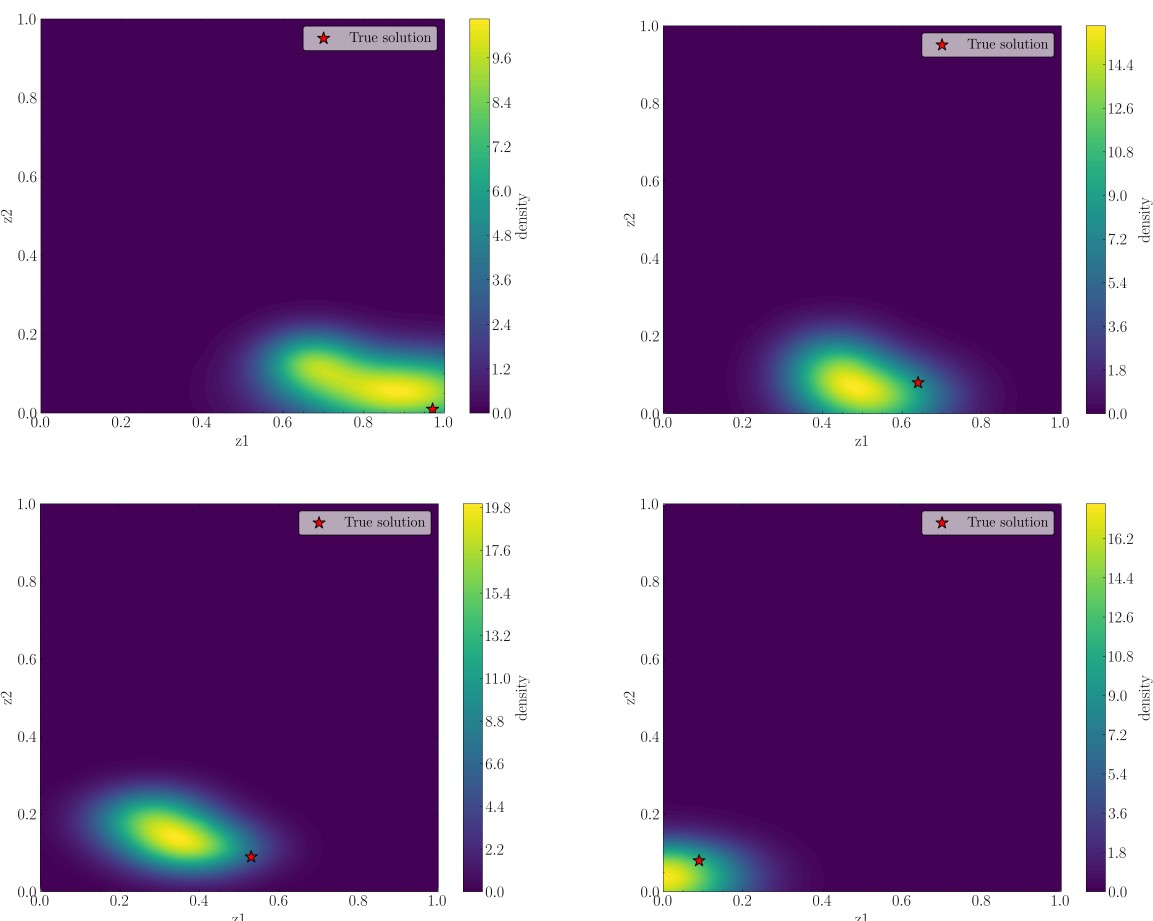

**Figure C2.** Posterior PDF ($q_{\boldsymbol{\theta}}(\mathbf{z}|\mathbf{m}, \mathbf{w})$) contour plots for four test scenarios when training with biofouling ($z_2$) scenarios $\geq 20\%$