# Peer review of "Gaussian Mixture autoencoder for uncertainty-aware damage identification in a Floating Offshore Wind Turbine"

_Wind Energy Science, 2024_

## Community Comment (CC1)

Response to Anonymous Referee Interactive Discussion

The authors would like to acknowledge the referee for the insightful and constructive comments regarding the manuscript. In the following, we answer to the comments and questions.

Scientific comments

1. In many machine-learning contexts (e.g., VAEs or standard autoencoders), the forward (decoder) and inverse (encoder) are trained jointly with a single objective. Clarify why two-step training is chosen over a single integrated approach, and discuss potential pros/cons.

In most autoencoders, the encoder and decoder must be found with a common goal. In contrast, herein, the decoder is dictated by a physical law and, therefore, it is fixed. We pre-train a Neural Network to approximate that physical law (decoder) for computational purposes since this enables access to its derivatives needed when training the encoder. Other than that, the decoder should follow the physical law and, therefore, it must be fixed. By having fewer unknowns (only those corresponding to the encoder) we minimize the problem difficulty; in particular, we decrease the number of local minima.

2. In the proposed method, the training of the forward operator is deterministic. Have you considered a probabilistic (or noisy) surrogate as well?

No, because the physical law governing the system is deterministic. The data employed to train the model incorporates noisy responses to account for measurement error. Since this work aims to explore how the uncertainty transfers from the measured observations to the estimated damage properties, we have neglected analyzing the uncertainty in the forward model.

3. Equation 9 needs to be clarified. What are the assumptions for the prior p(z)?

We assume uniform prior for the damage condition properties z1 and z2, constrained to the domain [a, b], as indicated in line 244 in the manuscript draft.

4. "Substituting Eq 9 in Eq 12 gives Eq 13". This part needs more detailed explanation or derivation steps for better comprehension.

Thank you for the observation. We will provide a more detailed derivation of the expression.

5. What are the features used for the measurements? Statistics of the time series, properties of the PSD?

In section 3, we describe the specifications of the case study, indicating the selected features employed as the measurements. We employ five features, including time series statistics and frequency-domain features from acceleration signals: the mean, the standard deviation, two dominant peak frequencies, and the zero-th momentum. Exploring Cross Power Spectral Density (CPSD) features will be considered in future work.

6. What are the architecture and training parameters of the deterministic counterpart?

We have omitted the details related to the decoder (forward operator) as it was optimized in a previous work cited in this work. We will incorporate an appendix to summarize this information.

7. Does the provided uncertainty represent aleatory, epistemic components or mixed? In the latter case, how to decompose it?

This work focuses on exploring the aleatoric uncertainty, assuming that the available training data sufficiently covers all the potentially observed scenarios. As it is a proof of concept, we have selected frequently occurring damages and assume that no different damage may occur. However, we consider as future work handling this uncertainty.

8. The study focuses on two specific damage types within a single mooring line. This constraint simplifies the problem but may not represent the diversity of real-world conditions, where multiple damage types may occur at various locations.

We appreciate the comment and agree with the limitations of the proposed work. However, as a proof of concept, we consider it mandatory to constrain the problem. The lack of experimental data from damaged turbines requires generating computationally expensive synthetic measurements, forcing us to constrain the problem for practical reasons.

9. Damage data and measurement data will be needed to train the forward operator in a supervised manner. Similar to the above comment, it works in the case study because the same damage mode is simulated for both training and testing. But in reality, it wil not be the case. In this context, how would you address the following barriers for practical application:
    1. Damage data can be rarely collected from the real structure and therefore, simulations will be required to train the model. The difference between the simulated response and the real turbine response will affect the model robustness.

The authors agree that this issue is one of the key barriers to SHM applications. Incorporating a calibration task according to a set of available measurements enables the reduction of the discrepancy between the real and the simulated domains. Performing domain adaptation techniques to take advantage of limited raw experimental data and enhance synthetic measurements is a key challenge to overcome this limitation, which is a future research line for us. However, it depends upon the availability of real data.

    2. The simulated dataset will not cover all possible damage scenarios.

We agree that enlarging the dataset to sweep a wide range of possible damage scenarios is challenging (experimental data scarcity and computational cost). Thus, we must prioritize the most frequently occurring damage cases according to the knowledge from experts in the field and the experience of aging or long-term instrumented systems. Once we gain access to experimental data from operating systems, we will incorporate more cases prone to occur during service in future works.

Technical comments

Literature review should be more structured.

We will review the literature review organization.

Citation style should be proper and consistent throughout the manuscript. Cite in parentheses wherever is relevant.

We will review the citation style.

The manuscript still needs careful and thorough proof-reading.

We will review the manuscript writing to look for errors.

Thank you.

Sincerely,

A. Fernandez-Navamuel, N. Gorostidi, D. Pardo, and V. Nava.

---

## Author Comment (AC1)

**Response letter WES-2024-160**

February 4, 2025

**From**: A. Fernandez-Navamuel, N. Gorostidi, D. Pardo, V. Nava, and Eleni Chatzi

**To**: Wind Energy Science journal editorial board

**Re**: Response Letter to Reviewers' comments (wes-2024-160)

We would like to acknowledge all the reviewers for their effort in reviewing our manuscript and providing very insightful and constructive comments. In the following, In the following, we answer to all received comments and queries.

**Response to RC1: "Comment on wes-2024-160":**

**Scientific Comments**

**SC1:** In many machine-learning contexts (e.g., VAEs or standard autoencoders), the forward (decoder) and inverse (encoder) are trained jointly with a single objective. Clarify why two-step training is chosen over a single integrated approach, and discuss potential pros/cons.

**Response:** As indeed stated by the reviewer, in most autoencoder formulations, the encoder and decoder must be found with a common goal. In contrast, herein, the decoder is dictated by a physical law and, therefore, is assumed fixed. This distinguishes the proposed approach from the common frameworks adopted in the literature. We pre-train a Neural Network to approximate the physical law (decoder) since this enables access to the derivative quantities that are needed when training the encoder. Moreover, the decoder serves to impose the known physical law as a type of inductive bias. A further benefit of this approach is that by specifying fewer unknowns (only those corresponding to the encoder), we reduce the difficulty of the inference task; in particular, we decrease the number of local minima.

We have included an extra paragraph in Section 4 to specify the benefits of this approach: "*We follow the two-step training procedure described in Section 2. We first find the optimal forward operator $\mathcal{F}_{\phi^*}$ by minimizing Eq.3, as described in our previous work [1]. We pre-train the decoder to approximate the physical law of the system, since this enables access to the derivative quantities that are needed when training the encoder, according to the loss function in Eq.16. In this manner, the decoder serves to impose the known physical law as a type of inductive bias. A further benefit of this approach is that by specifying fewer unknowns (only those corresponding to the encoder), we reduce the difficulty of the inference task; in particular, we decrease the number of local minima.*"

Moreover, in response to the query by RC2, we have trained the full VAE architecture in a single step, obtaining unsuccessful results compared to our approach. We have included some figures reporting these results in this letter (comment 1 in RC2).

**SC2:** In the proposed method, the training of the forward operator is deterministic. Have you considered a probabilistic (or noisy) surrogate as well?

**Response:** In our architecture, the layers of the encoder and the decoder are deterministic. However, we employ a probabilistic model (Gaussian Mixture) to describe the latent space (i.e., the damage condition). The probabilistic model reflects the effect of noise in the input signals (measurement error and slight modeling error) and the multimodality of the solution, given the ill-posedness of the inverse formulation. This architecture can be seen as a Variational Autoencoder (VAE) that accounts for uncertainty in the damage condition estimate. Considering larger modeling error effects (epistemic uncertainty) is part of our future work, as we now state in Section 7: "*Also, to achieve a higher level assessment, we will investigate clustering techniques to separate the different types of uncertainty according to the nature and properties of the*

*existing sources (e.g., multimodality of the estimate, measurement noise, model uncertainties, extrapolation over unknown measurements). [2]"*

**SC3:** Equation 9 needs to be clarified. What are the assumptions for the prior p(z)?

**Response:** We have clarified Eq. 9 indicating that p(z) refers to the prior distribution of the damaged condition properties, which is unknown. In the revised manuscript, we specify the assumptions for the prior $p(z)$ as a uniform distribution with bounds $[b_{low}, b_{up}]$ constraining the domain after Eq. 15: *"We then assume the noise follows a Gaussian distribution, $p(\boldsymbol{\epsilon}) = \mathcal{N}(0, \Gamma)$, where $\Gamma = diag(\beta \mathcal{F}([\mathbf{z}, \mathbf{w}]))^2$ is a vector that contains the non-zero elements of a diagonal matrix, and the parameter $\beta$ corresponds to the noise level. This allows us to rewrite Eq.6 as:*

$$p(\mathbf{z}|\mathbf{m}, \mathbf{w}) \propto p(\mathbf{m} - \mathcal{F}(\mathbf{z}, \mathbf{w})) \cdot p(\mathbf{z}) = \frac{1}{(2\pi)^{M/2}|\Gamma|^{1/2}} exp\left(-\frac{1}{2}(\mathbf{m} - \mathcal{F}([\mathbf{z}, \mathbf{w}]))^t \Gamma^{-1}(\mathbf{m} - \mathcal{F}([\mathbf{z}, \mathbf{w}]))\right),$$
(1)

*where $p(\mathbf{z})$ is the prior distribution of the damaged condition properties, which can follow any PDF."*

In the revised manuscript, we specify the assumptions for the prior p(z) as a uniform distribution with bounds $[\mathbf{b}_{low}, \mathbf{b}_{up}]$ constraining the domain, after Eq. 15: *"The second term refers to the prior, which we assume to follow a bounded uniform distribution $p(\mathbf{z}) \sim \mathcal{U}[\mathbf{b}_{low}, \mathbf{b}_{up}]$ with lower and upper bounds $\mathbf{b}_{low}$ and $\mathbf{b}_{up}$, respectively."*

**SC4:** Substituting Eq 9 in Eq 12 gives Eq 13". This part needs more detailed explanation or derivation steps for better comprehension.

**Response:** We now provide a more detailed derivation of the ELBO loss expression (Eqs. 11 to 16 in the revised manuscript): *"However, the KL divergence term exhibits certain shortcomings that weaken its strength as a distance metric; it is asymmetric, it does not satisfy the triangle inequality, and it produces an intractable term (the evidence of the data distribution $p(\mathbf{m})$) ([3]). Instead, a lower bound is calculated for the evidence, known as the Evidence Lower BOund (ELBO) ([3]). ELBO is the loss function commonly employed in Variational Autoencoders (VAEs) to account for the discrepancy between the two distributions ([4, 5]). We obtain the ELBO loss by exploiting the KL expression and isolating the intractable terms:*

[revised manuscript text omitted]

"

**SC5:** What are the features used for the measurements? Statistics of the time series, properties of the PSD?

**Response:** In section 3, we describe the specifications of the case study, indicating the selected features employed as the measurements. We employ five features, including time series statistics and frequency-domain features from acceleration signals: the mean, the standard deviation, two dominant peak frequencies, and the zero-th momentum. The expressions to calculate these features are introduced in Eqs. 17 to 22.

**SC6:** What are the architecture and training parameters of the deterministic counterpart?

**Response:** In Section 4 Table 1, we include the architecture and training parameters employed to train the decoder (deterministic part) during the first step of the training strategy. Figure 6 shows the evolution of the loss during this training step. *"We employ TensorFlow 2.13 to treat the datasets and train the Bayesian DNN for damage condition assessment ([6]). We split our dataset into training (*$\mathcal{D}_{train}$*), validation (*$\mathcal{D}_{val}$*), and testing (*$\mathcal{D}_{test}$*), each containing 70, 20, and 10% of the total samples, respectively. We then use the MinMax scaler ([7]) to constrain the environmental conditions and response features to the interval* $[0, 1]$ *so as to ensure that high-order features do not outweigh lower-order ones in the training process. The rescaling function is based on the training data and applies to the three datasets.*

*Next, we specify the DNN architecture and the hyperparameters describing the training routine. There are no general guidelines regulating the pursuit of an optimal configuration. Numerous hyperparameters exist in this case, pertaining to the model definition and its training*

*process, including layer counts, types, and sizes, activation functions, and regularization techniques, among others. This renders the search space virtually infinite. As a common practice, developers usually test the performance of a range of candidate architectures designed on educated guesses and experience, aiming to strike a balance between computational efficiency and prediction accuracy.*

*In this work, we employ a combination of hyperbolic tangent ([8]) and Rectified Linear Unit (ReLU) ([9]) functions for the hidden layers. We implement the weight initialization method proposed by Aldirany et al. ([10]), who suggested TensorFlow's default Glorot Uniform initialization scheme ([11]) to be unsuitable for non-differentiable and non-zero mean functions, such as ReLU. Instead, we apply the He Uniform initialization ([12]) to the ReLU layers. At the output layer, we use three different activations for the properties of the multivariate Gaussian Mixture: the sigmoid function ([13]) for the means, as we seek for a smooth function to estimate damaged condition coefficients in the interval $[0, 1]$; the softplus function ([14]) for the variances, as a smooth equivalent of ReLU to enforce positive values; and the softmax function ([15]) for the weights, so that their sum is equal to one and each value ranges into $[0, 1]$. We have observed adequate training performance when using the parameters shown in Table 2".*

| Encoder | |
|---|---|
| Layers | 100, 250, 300, 300, 200, 150, 100 |
| Activations | ReLU, ReLU, Tanh, ReLU, Tanh, ReLU, Tanh |
| Weight init. | GU, GU, HU, GU, HU, GU, HU |
| Initial LR | $10^{-5}$ |
| Batch size | 1024 |
| Epochs | 200 |
| **Sampling layer** | |
| $\mu$ activation | Sigmoid |
| $\sigma$ activation | Softplus |
| $w$ activation | Softmax |
| Num. Gaussians | 5 |
| Num. samples | 10 |
| **Decoder** | |
| Layers | 10, 30, 50, 70, 80 |
| Activations | Tanh, ReLU, ReLU, ReLU, ReLU |
| Weight init. | GU, HU, HU, HU, HU |
| Initial LR | $5 \cdot 10^{-3}$ |
| Batch size | 512 |
| Epochs | 500 |

Table 1: Specifications of our Gaussian Mixture autoencoder. Tanh: Hyperbolic Tangent; ReLU: Rectified Linear Unit; GU: Glorot Uniform, HU: He Uniform.

**SC7** Does the provided uncertainty represent aleatory, epistemic components or mixed? In the latter case, how to decompose it?

**Response:** This work explores uncertainty related to the multimodality of the estimated damage condition according to the observed inputs. This source of uncertainty is inherent to the ill-posedness of the inverse problem; various damage conditions may often produce the same measurement result (especially under sensor sparsity).

We further consider the aleatory and epistemic uncertainty that transfers from the measured signals (affected by added noise to represent sensor imprecision and light modeling error) to

the damaged condition space. However, this strategy does permit accounting for severe model mismatch (which offsets the mean of the estimate) and, thus, neglects a complete treatment of the effect of epistemic uncertainty. As it is a proof of concept, we have designed a methodology to detect damage scenarios that are similar to those employed during training. In this sense, we have selected frequently occurring damage types according to our expertise in the field and the literature review.

**SC8:** The study focuses on two specific damage types within a single mooring line. This constraint simplifies the problem but may not represent the diversity of real-world conditions, where multiple damage types may occur at various locations.

**Response:** We appreciate the comment and agree with the limitations of the proposed work. However, as a proof of concept, we consider it mandatory to constrain the problem. The lack of experimental data from damaged turbines requires generating computationally expensive synthetic measurements, forcing us to constrain the problem for practical reasons. We have included a more detailed explanation of the limitations of this work in the Conclusions.

To prove the applicability of the proposed method, we have considered an additional dataset where simultaneous damage – anchoring type – occurs in two mooring lines. We include the results in Appendix B, where we observe a successful identification when the three rotation DOFs are considered. However, when using only roll DOF, the damage in the second line turns undetectable. We consider future work (indicated in Section 7) increasing the dimensionality of the damage condition space (multiple-type damage occurring simultaneously at different assets, such as mooring lines).

**SC9:** Damage data and measurement data will be needed to train the forward operator in a supervised manner. Similar to the above comment, it works in the case study because the same damage mode is simulated for both training and testing. But in reality, it will not be the case. In this context, how would you address the following barriers for practical application:

**C9.a)** Damage data can be rarely collected from the real structure and therefore, simulations will be required to train the model. The difference between the simulated response and the real turbine response will affect the model robustness.

**Response:** The authors agree that this issue is one of the key barriers to SHM applications. In this context, the scarcity of real damage data needs to be complemented with simulation models that produce synthetic data as part of the training process. Making the simulation responses more reliable and close to field observations requires incorporating a calibration task according to a set of available measurements to allow for the reduction of the discrepancy between the real and simulated domains. Domain adaptation techniques can be implemented in this context, to take advantage of limited raw experimental data and refine synthetic data generation, we identify as a future research line.

**SC9.b)** The simulated dataset will not cover all possible damage scenarios.

**Response:** The authors agree that enlarging the dataset to sweep a wide range of possible damage scenarios is challenging (experimental data scarcity and computational cost) and further unrealistic.

However, we can demonstrate that the method is still valid when we evaluate scenarios that maintain the nature of the simulations used during training but remain unseen during the training phase. For this purpose, we have performed two different analyses for the inverse: (i) training only with scenarios where damage $z_1$ (anchoring) is always $\geq 20\%$ and testing with scenarios with $z_1 < 20\%$, and (ii) training only with scenarios where damage $z_2$ (biofouling) is always $\geq 20\%$ and testing with scenarios with $z_2 < 20\%$. In both cases, we consider using the three rotation DOFs as the input measured signals (roll, pitch, and yaw).

Figure 1 shows four randomly selected test cases for the first analysis (anchoring kept $\geq 20\%$

during training). We observe that the distributional model successfully approximates the true

[Figure]

Figure 1: Posterior PDF ($q_{\boldsymbol{\theta}}(\mathbf{z}|\mathbf{m}, \mathbf{w})$) contour plots for four test scenarios when training with anchoring ($z_1$) scenarios $\geq 20\%$

solution, with the uncertainty spreading mostly in the direction of the constrained feature (i.e., anchoring) within acceptable thresholds. Hence, the developed method can inform on the damage condition and indicate the uncertainty direction even for cases where the evaluated scenario lies outside the training domain.

Analogously, Figure 2 shows four test scenarios for the second analysis (biofouling kept $\geq 20\%$ during training). The results reveal that the highest density values concentrate close to the true solution, although with some error (see, e.g., Figure **??**, where the highest PDF values are shifted towards smaller anchoring damage, but the level of biofouling damage is well captured). In summary, the results demonstrate that despite these not being included during the training phase, the DNN successfully identifies damage scenarios with a lighter severity. The analyses demonstrate the generalizability of the method for damage scenarios that have the same nature but are significantly different in severity from those considered during training, as well as the potential of the distributional model to indicate the direction of the uncertainty. We now include in Appendix C the results of these analyses.

**Technical Comments**

**TC1:** Literature review should be more structured.

**Response:** We have revised the literature review for a better organization.

**TC2:** Citation style should be proper and consistent throughout the manuscript. Cite in parentheses wherever is relevant.

**Response:** We have revised the citation style for consistency throughout the manuscript.

[Figure]

Figure 2: Posterior PDF ($q_{\boldsymbol{\theta}}(\mathbf{z}|\mathbf{m}, \mathbf{w})$) contour plots for four test scenarios when training with biofouling ($z_2$) scenarios $\geq 20\%$

**TC3:** The manuscript still needs careful and thorough proof-reading.

**Response:** We have revised the manuscript to correct writing errors.

**Response to RC2: "Comment on wes-2024-160":**

**C1:** The authors combine a GMM with Autoencoders (using a deterministic encoder) to leverage the strengths of both methods, particularly for capturing multimodal latent representations. However, the paper should include a comparison between the proposed GMM-AE approach and a standard Variational Autoencoder for at least one case scenario. This would allow for evaluating reconstruction accuracy, clustering performance, and trade-offs such as increased model complexity while exploring whether the VAE's inherent stochasticity in the encoder provides comparable benefits.

**Response:** In most autoencoders, the encoder and decoder must be determined using a common target. Here, we employ deterministic layers to describe both the encoder (inverse) and the decoder (forward) and incorporate the effect of uncertainty via a distributional model that describes the latent space. The probabilistic model reflects the effect of noise in the input signals (measurement error and slight modeling error) and the multimodality given the ill-posedness of the inverse formulation. This architecture can be seen as a Variational Autoencoder (VAE) that accounts for uncertainty in the damage condition estimate. Regarding the training, we first train the decoder to approximate the physical laws that describe the forward (decoder) for computational purposes since this enables access to its derivatives needed when training the encoder. By having fewer unknowns (only those corresponding to the encoder), we minimize the problem difficulty; in particular, we decrease the number of local minima.

Since the reviewer's request is indeed valid, we have trained the entire VAE in a single step, obtaining unsuccessful results. Upon running an extensive ablation study for configuring the training hyperparameters (learning rate, optimizer, epochs), the authors found no way to replicate the results obtained with the two-step training approach. We include the evolution of the two-loss function terms during training in Figures 1 and 2 and the predicted contour plot for the first test case (Figure 3).

[Figure]

(a) Data misfit        (b) Mixture misfit        (c) Contourplot test case 1

Figure 3: Summary of results obtained when training the entire VAE architecture in a single step.

**C2:** The number of Gaussian components in a GMM is typically a hyperparameter that must be set before training. The authors set it equal to 5; why? there is no comment about this. Choosing the wrong number of components can lead to underfitting or overfitting.

**Response:** We have no prior knowledge of the optimal number of Gaussian components to describe the uncertainty of the damage condition properties. The larger the number, the more flexible the mixture will be. However, the number of components directly affects the number of parameters to be estimated; thus, it hampers the training process. After a trial and error analysis, the authors reached adequate results using five Gaussian components. Using more components than needed to accommodate the distributional shapes results in components with very low weight values to neglect their contribution. We now include an explanation in the manuscript in Section 4: "*For the Multivariate Gaussian Mixture PDF, we assign $k = 5$ Gaussian components. We have no prior knowledge of the optimal number of Gaussians to reflect the uncertainty of the damage condition properties. The larger the number, the more flexible the mixture will be. However, the number of Gaussians directly affects the number of parameters to be estimated; thus, it hampers the training process. After some trial and error analysis, the authors reached adequate results using five Gaussians.*"

**C3:** The authors need to properly discuss the limitations of using GMM.
    **C3a)** GMM suffers in the context of high-dimensional data. In this case, the damaged feature space is pretty small, which is convenient. How would this scale up?

**Response:** We thank the reviewer for this comment. We now clarify in more detail the limitations of the proposed method, particularly Gaussian Mixtures, in Section 1: "*Despite the successful results, this work suffers certain limitations that ought to be acknowledged. First, the methodology provides a way to describe the uncertainty that is inherent to the ill-posedness of the inverse problem. Such uncertainty induces the multimodality of the output (i.e., the damage condition estimate). The method also reflects the effect of aleatory uncertainty (noisy measurements) as it transfers from the measured data to the estimated condition. However, this work neglects the epistemic uncertainty, which occurs when making a prediction on measurements that correspond to a damage condition that is far from those employed in the training stage. Accounting for such uncertainty and disentangling both sources is beyond the scope of this work and requires further study. Second, since the Gaussian mixture is a parametric approach, it*

*constrains the outcome and prevents a complete characterization of the effect of uncertainty. Improving the distributional model requires many components, which enormously increases the number of parameters to be estimated. This is an important limitation of the method, mostly when scaling up to higher-dimensional spaces. Finally, due to the current scarcity of experimental data from real operating FOWTs, this work is entirely restricted to synthetic data from computational simulations. Integrating experimental data with synthetic scenarios is a key challenge to proving the applicability of the suggested methodology in real-field data. "*

Our methodology employs a Variational Autoencoder (VAE) with a two-step training strategy to account for uncertainty in damage identification from measurements. In this framework, a Gaussian Mixture is chosen for modeling the latent space due to its ability to describe multimodal distributions effectively, particularly in low-dimensional settings. However, we acknowledge that GMMs may struggle in high-dimensional spaces, where their capacity to capture complex dependencies and multimodal structures diminishes. To address this limitation, we mention in the revised text that alternative distributional models, such as Copulas, may offer improved scalability and flexibility for higher-dimensional damage condition spaces. Additionally, as part of our ongoing and future work, we are actively investigating the comparative performance of GMMs and Copulas in handling increased dimensionality. This will provide deeper insights into the robustness of our approach as the complexity of the problem grows.

**C3b)** GMM assumes that the data can be modeled as a combination of Gaussians. What if this assumption is not met?

**Response:** Given that there is no prior knowledge about the distributional shape of the data in the latent space, we look for a sufficiently flexible PDF. Here, we use a mixture of Gaussians to reflect the multimodality of the solution space, which, with sufficient components, can accommodate complex shapes with a relatively reduced model complexity. However, exploring other distributional models is part of our future research work, including using Copula functions, although these may encounter more difficulties in describing multimodality. We now indicate this future line in Section 7: *"Scaling up the methodology to more complex damage spaces (e.g., simultaneous damage in all mooring lines, different directions of the anchor displacements, corrosion effect) is one of the key challenges to be addressed. Given the intractable dimensionality increase in the number of parameters of the Gaussian mixture, we will explore alternate distributional models, such as random fields ([16]), Gaussian Processes (GPs) ([17]), or Copulas and vine Copulas ([18])."*

**C3c)** The EM algorithm behind GMM is sensitive to initialization - how is it initialized here?

**Response** We select layer-specific weight initialization methods depending on the nonlinear function, according to references in the field. We describe and justify these decisions in section 4, including the selected initialization methods in Table 1. With these specifications, the authors observed adequate behavior of the loss function during the training process: *"In this work, we employ a combination of hyperbolic tangent ([8]) and Rectified Linear Unit (ReLU) ([9]) functions for the hidden layers. We implement the weight initialization method proposed by Aldirany et al. ([10]), who suggested TensorFlow's default Glorot Uniform initialization scheme ([11]) to be unsuitable for non-differentiable and non-zero mean functions, such as ReLU. Instead, we apply the He Uniform initialization ([12]) to the ReLU layers. At the output layer, we use three different activations for the properties of the multivariate Gaussian Mixture: the sigmoid function ([13]) for the means, as we seek for a smooth function to estimate damaged condition coefficients in the interval $[0, 1]$; the softplus function ([14]) for the variances, as a smooth equivalent of ReLU to enforce positive values; and the softmax function ([15]) for the weights, so that their sum is equal to one and each value ranges into $[0, 1]$. We have observed adequate training performance when using the parameters shown in Table 2"*.

| Encoder | |
|---|---|
| Layers | 100, 250, 300, 300, 200, 150, 100 |
| Activations | ReLU, ReLU, Tanh, ReLU, Tanh, ReLU, Tanh |
| Weight init. | GU, GU, HU, GU, HU, GU, HU |
| Initial LR | $10^{-5}$ |
| Batch size | 1024 |
| Epochs | 200 |
| **Sampling layer** | |
| $\mu$ activation | Sigmoid |
| $\sigma$ activation | Softplus |
| $w$ activation | Softmax |
| Num. Gaussians | 5 |
| Num. samples | 10 |
| **Decoder** | |
| Layers | 10, 30, 50, 70, 80 |
| Activations | Tanh, ReLU, ReLU, ReLU, ReLU |
| Weight init. | GU, HU, HU, HU, HU |
| Initial LR | $5 \cdot 10^{-3}$ |
| Batch size | 512 |
| Epochs | 500 |

Table 2: Specifications of our Gaussian Mixture autoencoder. Tanh: Hyperbolic Tangent; ReLU: Rectified Linear Unit; GU: Glorot Uniform, HU: He Uniform.

**C4:** The authors also use the operational loading conditions as input. You are never specific about which one you are using. You mention wave height, wind speed, and peak period for the simulations. Are those the input as well? How sensitive is your solution to having/not having those inputs to the encoder and then back to the decoder?

**Response:** We now describe the selected operational loading conditions used as input in more detail. The three considered features correspond to frequently measured variables in real operating systems that are often available in practice thanks to the Supervisory Control and Data Acquisition (SCADA) systems: "*The simulation process follows the same structure as that presented by Gorostidi et al. ([1]). We assign the environmental conditions for each simulation by selecting a combination for significant wave height $H_S \in [2, 15]$ (m) and peak period $T_P \in [1, 15]$ (s). The decision to use these features is motivated by their common availability in real practice, since Supervisory Control And Data Acquisition (SCADA) systems typically include such measurements. We have defined evenly-spaced values for both $H_S$ and $T_P$ within their feasible interval, and one combination is randomly selected for each simulation using Monte Carlo sampling. These two variables define a Pierson-Moskowitz spectrum, which estimates the distribution of the energy of ocean waves based on their frequency using the empirical correlation proposed by ([19]). The integration of this spectrum defines the temporal evolution of the wave force component of the total force in Equation 23. We select wind velocity $W_V$ in a similar manner, with speeds ranging from 1 to 30 m/s. In this work, we have considered uniform wind speed profiles.*"

Appendix A includes the results obtained when training the inverse operator Neural Network without the loading conditions as input. The results reveal that neglecting the loading conditions prevents adequately capturing the distributional model for some test scenarios.

**C5:** They should consider extending their analysis to scenarios that involve simultaneous damage to two mooring lines. This will allow them to test the capability of the methodology to capture the coupled dynamics between the lines and their impact on the floating platform's overall response.

**Response:** To analyze this problem, we have considered a new dataset with simultaneous damage that occurs in two different lines. For simplicity in the analysis, we have only considered one damage type (anchoring) for each line to remain in a two-dimensional space. Increasing the dimension brings additional complexity to the problem and demands extensive analysis that is out of the scope of this work. Appendix B summarizes this case study and shows the results obtained for some test scenarios, where we observe the ability of the method to detect simultaneous damage occurring in two mooring lines when the three rotation DOFs are considered as input (roll, pitch, and yaw).

**Response to EC1: 'Comment on wes-2024-160':**

Many thanks to the authors and reviewers for their efforts. A few additional comments:

**C1:** If I understand correctly, the authors consider mainly a case where each of their damage estimations is based on a single realization of the environmental conditions and the system response (which is itself based on time series aggregated over a period of time, say 10 minutes). This is also the logical approach with numerical simulations where realizations are statistically independent from each other. In real systems, the different variable categories (environmental conditions, response and damage variables, and noise/errors) will have varying degrees of autocorrelation. This property may be exploitable by doing further data aggregation, to increase the confidence in the results. Could the authors please comment/consider this possibility?

**Response:** In this work, we have considered only the time-domain variability of the loading conditions (wind and wave excitation) according to the Pierson-Moskowitz spectrum, which estimates the energy distribution of the ocean waves based on their frequency. However, at this preliminary stage, we have neglected the variability of phenomena that may affect the system's dynamics (e.g., the effect of temperature on the stiffness). This consideration increases the model's complexity and needs to be addressed in future work. Besides, exploring autocorrelation as an additional feature to exploit the information contained in the data will be considered in future work. We now include this research line in Section 7: "*We further consider a future research line to focus on more exhaustive feature extraction that includes more refined system identification features, including the transmissibility and transmittance, or autocorrelation features, as they may enhance the damage identification task.*"

**C2:** I could imagine this methodology being applicable to a broader range of problems. Do the authors agree, and do they see any particular challenges/limitations?

**Response:** The authors agree that the methodology is transferable to many problems related to Structural Health Monitoring (SHM). Currently, we are attempting to transfer the proposed methodology to identify damage in single-span bridges. In this field, one of the key challenges relates to the scalability of the methodology. Since bridges are large structures, the damage condition space comprises many variables (there may exist many damage conditions, types, and locations). The Gaussian Mixture model may fail in such complex domains. Hence, exploring more efficient distributional models (e.g., Gaussian Mixture Copulas with mixture marginals) in high-dimensional spaces is required. Another issue that we encounter relates to the availability of experimental data regarding Environmental and Operational Conditions (EOCs). Whereas in the field of wind turbines the SCADA system is frequently available, civil engineering systems often lack detailed measurements of the affecting EOCs (e.g., temperature, humidity, traffic). We now state this in Section 7: "*Finally, given the potential of the proposed*

*methodology to be applied to different inverse problems, the authors are currently exploring its application to the field of damage identification for bridge structures. The main challenges arising in this context relate to the high dimensionality of the damage condition space, which will require more efficient distributional models such as copulas."*